# Molecular basis and cellular functions of vinculin-actin directional catch bonding

Venkat R. Chirasani [1,8], Mohammad Ashhar I. Khan [1,8], Juilee N. Malavade [2,8], Nikolay V. Dokholyan [3,4,5] ✉, Brenton D. Hoffman [2,6] ✉ & Sharon L. Campbell [1,7] ✉

The ability of cells and tissues to respond differentially to mechanical forces applied in distinct directions is mediated by the ability of load-bearing proteins to preferentially maintain physical linkages in certain directions. However, the molecular basis and biological consequences of directional force-sensitive binding remain unclear. Vinculin (Vcn) is a load-bearing linker protein that exhibits directional catch bonding due to interactions between the Vcn tail domain (Vt) and filamentous (F)-actin. We developed a computational approach to predict Vcn residues involved in directional catch bonding and produced a set of associated Vcn variants with unaltered Vt structure, actin binding, or phospholipid interactions. Incorporation of the variants did not affect Vcn activation but reduced Vcn loading and altered exchange dynamics, consistent with the loss of directional catch bonding. Expression of Vcn variants perturbed the coordination of subcellular structures and cell migration, establishing key cellular functions for Vcn directional catch bonding.

The ability of mammalian cells to generate, resist, and respond to mechanical forces applied in distinct directions is critical to many fundamental biological processes, including the maintenance of cytoskeletal order, directed cell migration, and the development of anisotropic mechanical properties of tissue[1–4]. Traditionally, mechanical directional sensitivity has been attributed to F-actin, as it has an innately polar structure with distinct pointed and barbed ends and is often highly aligned in load-generating subcellular structures such as stress fibers[5,6]. However, to enable alteration or sensing of the mechanical nature of the cellular microenvironment, forces must be transmitted through cellular adhesion structures. Key examples include focal adhesions (FAs) and adherens junctions (AJs), which respectively mediate cell-extracellular matrix (ECM) and cell-cell interactions[7,8]. The overall organization of these subcellular structures is similar[9]. Both are composed of transmembrane receptors, integrins within FAs and cadherins within AJs, which are indirectly coupled to the force-generating actomyosin cytoskeleton through large protein plaques composed of many proteins[7] (Supplementary Fig. 1a, b). Specific multi-protein linkages that connect receptors and the actomyosin cytoskeleton have been identified in several cases[8,10]. Prominent examples include fibronectin (FN):integrin:talin:vinculin (Vcn):F-actin in FAs and E-cadherin:β-catenin:α-catenin:Vcn:F-actin in AJs. Thus, the ability of these multi-component linkages to maintain connectivity in response to directional mechanical forces is likely a primary, but poorly understood, determinant of both the bidirectional force transmission between the cell and its environment as well as the initiation of the mechanosensitive processes that regulate the assembly/disassembly of FAs and AJs[11].

The dynamic and load-bearing properties of multi-component mechanical linkages are primarily investigated by probing individual protein-protein interactions[12]. The effect of applied force on the strength of these interactions, often measured as bond lifetime, is used

[1]Department of Biochemistry & Biophysics, University of North Carolina at Chapel Hill, Chapel Hill, NC, USA. [2]Department of Biomedical Engineering, Duke University, Durham, NC, USA. [3]Department of Pharmacology, Penn State College of Medicine, Hershey, PA, USA. [4]Department of Biochemistry & Molecular Biology, Penn State College of Medicine, Hershey, PA, USA. [5]Department of Chemistry, Penn State College of Medicine, Hershey, PA, USA. [6]Department of Cell Biology, Duke University, Durham, NC, USA. [7]Lineberger Comprehensive Cancer Center, University of North Carolina at Chapel Hill, Chapel Hill, NC, USA. [8]These authors contributed equally: Venkat R. Chirasani, Mohammad Ashhar I. Khan, Juilee N. Malavade. ✉e-mail: dokh@psu.edu; brenton.hoffman@duke.edu; campbesl@med.unc.edu

to classify their responses. Outside the context of adhesion biology, most protein–protein interactions behave as slip bonds, which exhibit reduced binding strength/time with increasing applied force[13]. In contrast, many of the protein interactions within mechanical linkages in loading-bearing subcellular structures are catch bonds, exhibiting increased bond strength/lifetime with increasing applied force[14]. Examples of catch bonds include the interactions between FN:$\alpha_5\beta_1$ integrin, FN:$\alpha_V\beta_3$ integrin, talin:F-actin, Vcn:F-actin, the X-dimer configuration of E-cadherin:E-cadherin, alpha-catenin:F-actin, and F-actin:myosin[11,15–18] (Supplementary Fig. 1c). Furthermore, a substantial fraction of the interactions involving F-actin, including Vcn:F-actin, talin:F-actin, and alpha-catenin:F-actin, exhibit a directional dependence, with force applied toward the pointed end of the polar F-actin exhibiting greater catch bonding[11,17,18]. However, the mechanisms enabling directional catch bonding as well as the role of specific directionally-sensitive catch bonds in mediating subcellular structure organization and mechanosensitive cellular processes are poorly understood.

In this work, we explored the molecular basis and functions of directional catch bonding in the context of Vcn. Vcn is a highly expressed ubiquitous 117 kDa protein with three structural domains: a head domain (Vh), a proline-rich linker, and a tail domain (Vt)[19]. Vcn exhibits force-sensitive localization to FAs and AJs to regulate cell morphology, motility, and force transmission[20]. Knockout of Vcn in mice causes cardiac and neural tube developmental defects and embryonic lethality at ~ E10[21]. Mutations in Vcn are associated with muscle defects and cardiomyopathies[22]. Vcn-null murine embryonic fibroblasts (Vcn$^{-/-}$MEFs) exhibit spreading and adhesion defects, increased motility, and resistance to apoptosis and anoikis[23]. The tension across Vcn was shown to regulate a mechanosensitive switch governing the assembly/disassembly dynamics FAs[24]. Amongst the catch bonds characterized at a single interface, Vcn exhibits the most force-induced strengthening and substantial directional asymmetry (Supplementary Fig. 1c). Previous work has shown that Vt primarily mediates both actin binding and catch bonding[11], providing a simplified context to identify the determinants of directionally asymmetric catch bonding.

To probe the molecular mechanisms of Vcn catch bonding, we performed Discrete Molecular Dynamics (DMD) simulations of the F-actin:Vt complex subject to pulling in a variety of directions relative to F-actin. Constant pulling of Vt away from F-actin was executed in the pointed ($F_{pointed}$), barbed ($F_{barbed}$), and normal ($F_{normal}$) directions of F-actin. Umbrella sampling revealed that this simplified system captured key aspects of directional asymmetric Vcn:F-actin catch bond previously observed using single molecule approaches[11]. Analysis of intermolecular interactions during directional pulling simulations reveals specific residues involved in interactions that are selectively formed or strengthened in response to force applied toward the pointed end of F-actin. We term these directionally-asymmetric force-strengthening (DAFS) residues. To assess the effects of reduced directional catch bonding on Vcn function, we systematically designed and constructed a set of Vt and Vcn DAFS variants. Biophysical and biochemical characterization revealed that the Vt DAFS variants retain structure, actin binding, actin crosslinking, and lipid binding properties relative to WT. To assess the effects of DAFS on Vcn function in cells, fluorescent protein (FP)-tagged Vcn as well as FRET-based biosensors for Vcn conformation[25] and tension[26] harboring DAFS variants were created. DAFS variants did not exhibit perturbed FA assembly or Vcn conformation, but inclusion of increasing numbers of DAFS variants led to partial unloading of Vcn and alteration in the tension sensitivity of Vcn exchange dynamics, as expected for reduced catch bonding. Cells expressing the DAFS variant predicted to have the least catch bonding exhibit defects in the spatial distribution of loads within FAs aligned with F-actin, impaired coordination between FAs and the actomyosin cytoskeleton, and an inability to undergo haptotaxis in

Boyden chamber assays. Overall, this work informs the mechanistic understanding of directionally asymmetric catch bonding, elucidates the role of Vcn catch bonding in subcellular organization and cellular processes, as well as describes a suite of tools for studying Vcn bonding in a variety of contexts.

## Results

### Molecular dynamics simulations of Vt:F-actin complex replicate direction sensitivity of Vcn catch bonding

In full-length Vcn, autoinhibitory interactions between the head and tail domains (Vt) mask areas that mediate Vcn:F-actin interactions, making studies of actin interactions in the context of full-length Vcn challenging[19]. Therefore, the isolated Vt domain (881-1066) of Vcn is widely used to model Vcn:F-actin interactions for in silico and in vitro studies[27,28]. Previous in vitro work used single-molecule force spectroscopy (SMFS) to show that the Vt:F-actin complex exhibits directionally sensitive catch bonding, with interactions maximally stabilized by forces applied toward the $F_{pointed}$ direction[11]. We sought to better understand the energetics of these directionally asymmetric interactions. To do so, we employed umbrella sampling, which introduces an external biasing potential along a chosen reaction coordinate or order parameter to readily explore complex free energy landscapes within reasonable simulation times (Please refer to the Discussion section for a more in-depth discussion regarding the application of this approach to the study of catch bonding). We determined the free energy surfaces along chosen directions $F_{pointed}$, $F_{barbed}$, and $F_{normal}$ (Fig. 1a), and generated one-dimensional potential of mean force (PMF) curves from 26 sampling windows to obtain directional interaction energies between F-actin and Vt. Interaction energies between F-actin and Vt were dependent on the direction of the force relative to the polarity of F-actin ($\Delta G_{pointed}$ = 54.73 kcal/mol > $\Delta G_{barbed}$ = 33.09 kcal/mol > $\Delta G_{normal}$ = 25.02 kcal/mol, Fig. 1b). Notably, these computational results correlate well with SMFS studies probing the relative strength of directional catch bonding between Vt and F-actin[11], validating the overall computational approach for studying this interaction.

### Vt forms directionally asymmetric force strengthening (DAFS) H-bonds with F-actin

In the cryo-EM reconstruction of the Vt:F-actin complex[29], Vt residues in helix-4 and helix-5 bind to two adjacent actin subunits and constitute the primary unloaded binding interface. Individual subunits within the F-actin filament are referred to as protomer-P, corresponding to the subunits located toward the pointed end, and protomer-B, corresponding to the subunits located toward the barbed end of the F-actin filament. This interface has been validated using mutagenesis and F-actin binding analyses[29], and a few mutants have been shown to greatly reduce F-actin binding, including the Vt I997A mutation in helix-4[30]. Residue-level contact analysis of the Vt:F-actin cryo-EM reconstruction (PDB ID:3JBI [https://doi.org/10.2210/pdb3jbi/pdb] (vinculin tail domain bound to F-actin))[29], identified Vt residues (helix-4; R976, N980, Q983, R987, R1008 and helix-5; Q1018, N1026, R1039, E1040, and E1042) involved in H-bond interactions with F-actin residues[29] (Fig. 2a). These native interactions were retained in DMD simulations of Vt:F-actin at 0 pN force (unloaded state), which stipulate further probing of the force sensitivity and directional dependence of Vt:F-actin interactions.

To elucidate the properties and atomistic details of DAFS interactions between Vt and F-actin, we performed constant-force pulling DMD simulations[31] on the Vt:F-actin complex (PDB ID:3JBI)[29]. The complex was loaded by harmonically restraining the position of F-actin and applying a discretized step function with constant energy jumps at equal distance intervals to the complex in one of three directions. We selected the center of mass (CoM) of Vt as the reference point for applying external forces. To broadly characterize the system, a large set of pulling simulations were performed with forces ranging from

0-150 pN in $F_{pointed}$, $F_{barbed}$ and $F_{normal}$ directions and with each condition repeated ten times for a total of 4530 pulling simulations. The simulation space was considered sufficiently probed, as the Vt:F-actin complex was first observed to dissociate in the range of 125–132 pN, with dissociation occurring rapidly at 150 pN with directional

asymmetry (Supplementary Movies 1–6). As such, detailed analyses focused on the window of 124–132 pN. While the force amplitudes applied in these simulations are distinct from values observed in live cells (~2.5 pN)[26], this is common in DMD simulations, as the forces must be increased to allow feasible computational time[32]. Notably, these

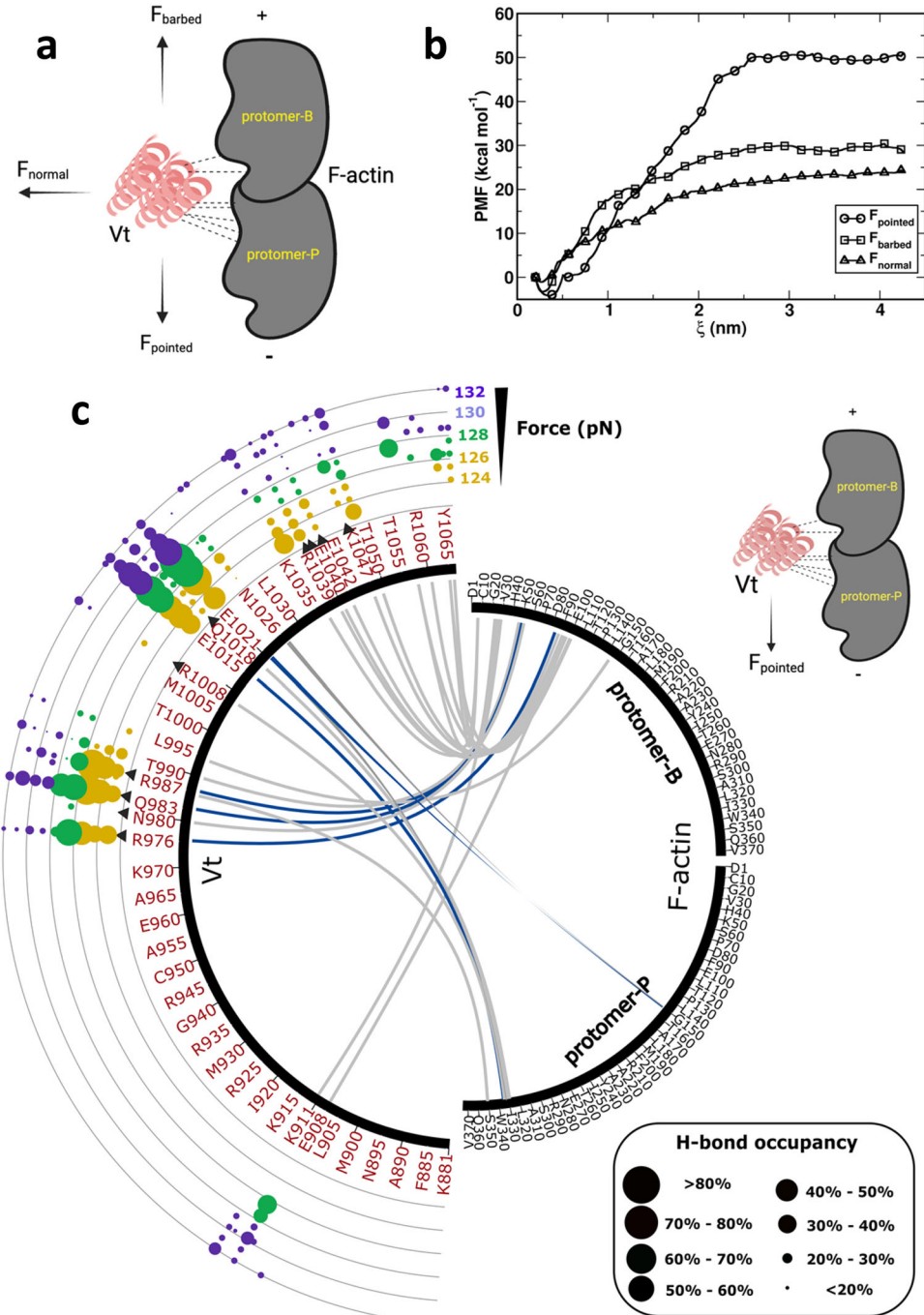

**Fig. 1 | Evaluation of energetics and hydrogen bond (H-bond) properties between F-actin and Vt. a** Schematic representation of pulling forces imposed on Vt in different directions during MD simulations. **b** Quantification of Vt:F-actin binding strength through potential of mean force (PMF) calculations. The X-axis represents the reaction coordinate (ξ), and the Y-axis represents the PMF. The graph displays PMF profiles for three pulling directions: along the $F_{pointed}$, $F_{barbed}$, and $F_{normal}$ directions, as Vt is pulled away from the actin filament. **c** A circular plot depicting Vt and F-actin interactions in the $F_{pointed}$ direction to probe Vt:F-actin engagement. The occupancy percentages of inter-molecular H-bonds between F-actin and Vt as quantified from constant-force pulling DMD simulations were

used to create the circular plot. A scatter plot was used to illustrate the occupancies of H-bonds, allowing for a detailed analysis of the H-bond occupancies at specific forces. Each dot on the plot represents the existence of H-bonds at different pulling forces, color-coded according to force magnitude. The size of each dot indicates the occupancy percentage of the corresponding H-bond during the pulling simulation at a specific force. Native H-bonds observed in the cryo-EM structure of the Vt:F-actin complex (PDB: 3JBI) are represented as black triangles. DAFS H-bonds that signify directional catch bonding between Vt and F-actin are illustrated as blue ribbons while normal H-bonds are represented by grey ribbons.

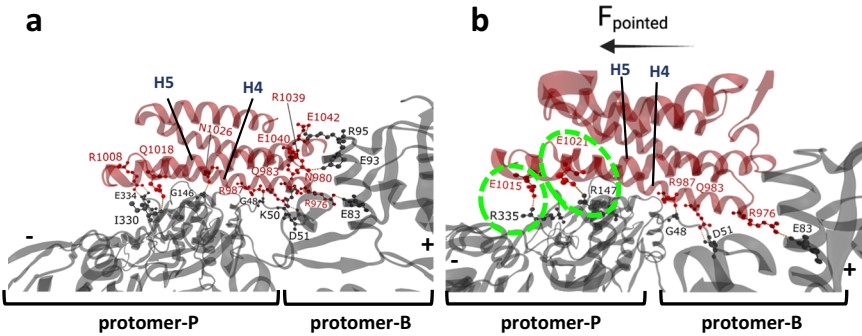

**Fig. 2 | Characterization of directionally asymmetric force-strengthening (DAFS) H-bond interactions between F-actin and Vt.** Close-up view of the Vt/F-actin interface (PDB ID: 3JBI) highlighting **a**, H-bonds between Vt and F-actin and **b**, DAFS interactions for the force-loaded state of Vt:F-actin complex in the $F_{pointed}$ direction. Newly induced DAFS interactions from $F_{pointed}$ trajectories are enclosed in dotted circles. Color scheme: Vt is shown in red and F-actin in gray. The residue pairs involved in H-bond interactions are shown in ball and stick representation.

conditions have been shown to reflect mechanosensitive processes within physiological force regimes[32,33].

Directional catch bonding is likely mediated by DAFS interactions that are weak or nonexistent at low forces and strengthen with increasing pulling force only in certain directions. To investigate the basis of F-actin and Vt catch bonding, we sought to broadly characterize the direction and force sensitivity of a variety of intermolecular interactions. Analysis of the Vt:F-actin contact surface area, number of salt bridges, and van der Waals (VdW) interactions revealed retention, but not strengthening, of these interactions to higher force magnitudes in the $F_{pointed}$ direction in comparison to $F_{normal}$ and $F_{barbed}$ (Supplementary Fig. 2). Apart from H-bond interactions, we have identified several hydrophobic (non-bonded) and salt-bridge interactions that play a significant role in mediating directional catch bonding between F-actin and Vt in the $F_{pointed}$ direction (Supplementary Table 21). The reinforcement of these interactions, facilitated by key force-induced H-bonds, contributes significantly to the establishment of unique catch bonding directionality. The combination of H-bond, salt-bridge, and non-bonded interactions is instrumental in establishing the unique directionality of catch bonding. Further, the salt-bridges and non-bonded interactions collectively enhance the stability of the Vt:F-actin complex under mechanical force. Our multifaceted approach in the current study underscores the complexity of catch bonding, emphasizing the importance of considering a range of interaction types to comprehensively understand the directional specificity observed in the Vcn:F-actin system.

Next, we assessed the effects of directional forces on H-bond formation by analyzing H-bond occupancy, which is defined as the percentage of DMD steps in a simulation where a specific H-bond is present, as a function of the applied force in different directions[18]. Consistent with directional catch bonding, an increase in the average number of H-bonds with increasing force was observed in the $F_{pointed}$ direction and not in $F_{barbed}$ or $F_{normal}$ directions (Supplementary Fig. 3a–c). Furthermore, analysis of native H-bonds revealed that the fraction of occupied bonds increased only when forces were applied in the $F_{pointed}$ direction, indicating DAFS for these specific interactions (Supplementary Fig. 3d).

To illustrate the relative H-bond occupancies of native and non-native H-bonds as a function of applied forces, circular plots were generated over the force regime that dissociation of the Vt:F-actin complex occurs (124-132 pN) (Fig. 1c and Supplementary Fig. 4). H-bonds with occupancies >=40% were considered long-lived, occupancies between 20% to 40% were considered medium duration, and occupancies <20% were considered short-lived interactions. H-bonds showing an increase or decrease in occupancy by a factor greater than 1.1 from their unloaded state or previous low force applied state were categorized as force strengthening or weakening, respectively.

Directional asymmetry was detected by comparing force-sensitive changes in occupancy percentages across different pulling directions.

In the $F_{pointed}$ direction, 35 H-bonds were detected between Vt and F-actin. Twenty-seven were short-lived, 3 were of medium duration, and 5 were long-lived. Out of 27 short-lived H-bonds, 7 were weakened with increased pulling force, and 20 were ruptured by 132 pN. All medium-duration interactions (E986/G48, T990/G46, and Q994/Y143, Vt/F-actin) were non-native interactions but substantially weakened with force. Interestingly, all five long-lived H-bonds were strengthened by force (125-128 pN) (Fig. 1c). Three of these long-lived H-bonds (R976/E83, Q983/D51, R987/G48 Vt/F-actin) are native interactions formed between helix-4 of Vt and F-actin protomer-B. The other two formed non-native interactions between helix-5 of Vt and F-actin and exhibited force strengthening. Specifically, E1015 interacts with R335 of protomer-P, while E1021 interacts with R147 of protomer-P. Beyond the pulling force of 128 pN, the DAFS interactions in the $F_{pointed}$ direction were weakened and eventually resulted in the separation of the Vt:F-actin complex. This is consistent with the expected catch-slip transition of realistic bonds[11,18].

During force application in the $F_{barbed}$ direction, 21 H-bonds were identified between Vt and F-actin. Eighteen were short-lived, three were medium-duration, and no long-lived interactions were detected. All three medium-duration interactions were native (R976/E83, R987/G48, R1039/E93 Vt/F-actin). The medium–duration interactions weakened with increased pulling force. Interestingly, five out of 18 short-lived H-bonds (Q983/D5, Q1018/A331, N1026/G146, E1040/K50, and E1042/R95 Vt/F-actin), are native interactions and marginally strengthened in the force range of 125–128 pN (Supplementary Fig. 4a). The remaining 13 short-lived interactions weakened with increased pulling force. Beyond the pulling force of 128 pN, all interactions displayed weakening until the rupture of the Vt:F-actin complex.

Loading in the $F_{normal}$ direction revealed 21 H-bond interactions. All these interactions were short-lived, with occupancy percentages below 20% (Supplementary Fig. 4b). Eight of the 21 were non-native H-bonds. All 21 short-lived interactions were weakened with increased pulling force. These findings are consistent with the rapid dissociation of the Vt:F-actin forces in the $F_{normal}$ direction (Supplementary Movies 1–6).

Altogether, these analyses reveal the atomistic interactions mediating directional asymmetry and force dependence of Vt:F-actin complex dissociation. Notably, different H-bonds were formed in various pulling directions (Fig. 1 and Supplementary Fig. 4). Maximal H-bond stabilization occurred with forces applied in the $F_{pointed}$ direction due to a conformational change in Vt at the Vt:F-actin interface that was not observed in the other directions. This stabilization is due to a combination of native and non-native H-bonds

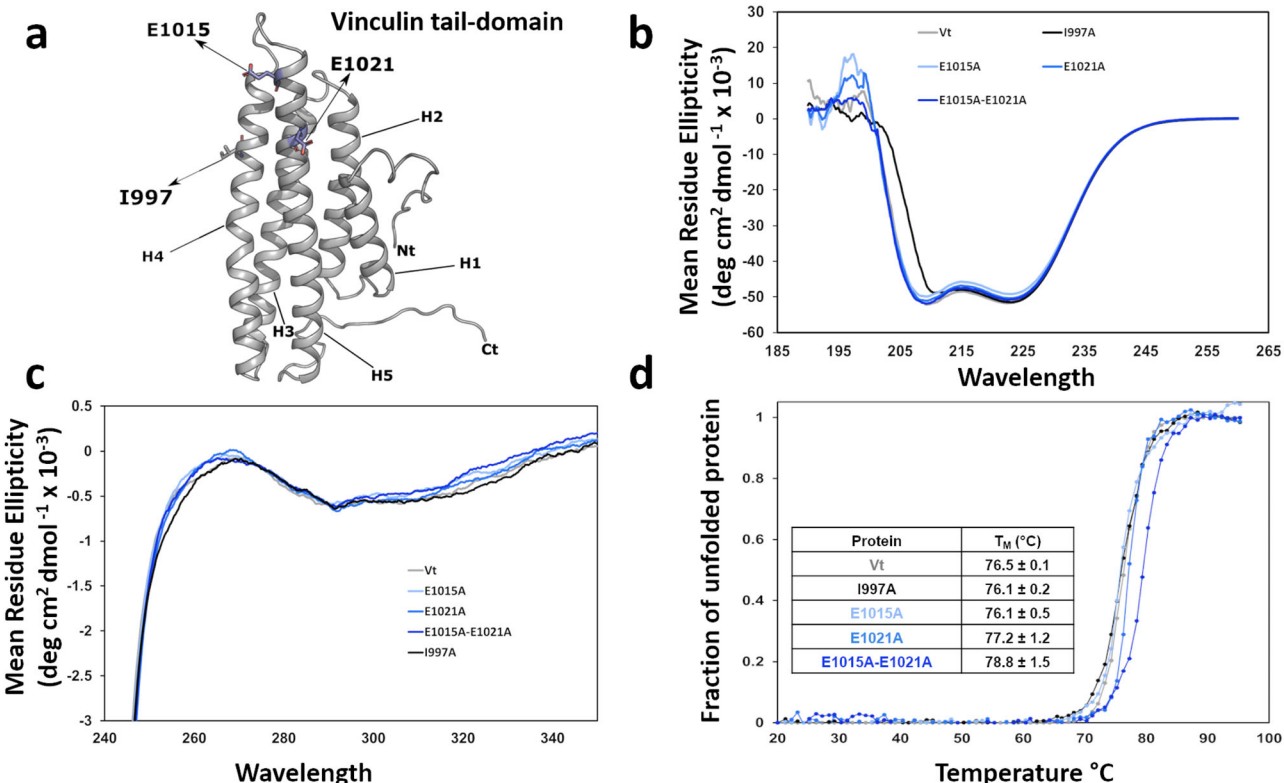

**Fig. 3 | DAFS residue variants retain Vt structure and stability. a** Ribbon diagram of Vt with residues predicted to form DAFS interactions highlighted in blue and a key actin binding residue (I997) in grey licorice. **b** Far-UV CD spectral profiles of DAFS variants. **c** Near-UV spectral overlay of DAFS variants. **d** Melting temperature

($T_M$) of DAFS variants calculated from CD thermal melt curves. Representative far-UV and near-UV CD spectral profiles shown are from $N = 3$ independent experiments. The CD thermal melt profiles are presented as an average of 3 scans from $N = 3$ independent experiments. Error in $T_M$ is shown with standard errors.

(Fig. 2). Interestingly, the long-lived, force-strengthening native H-bonds are induced by helix-4 residues of Vt. These include the H-bond interactions between the guanidino sidechain of R976 (Vt) and the side chain carboxyl group of E83 (protomer-B), the sidechain amino group of Q983 (Vt) and the side chain carboxyl group of D51 (protomer-B), and the sidechain amino group of R987 (Vt) and the backbone carboxyl group of G48 (protomer-B) (Fig. 2b). Furthermore, the long-lived, force-strengthening, non-native H-bonds are induced by helix-5 residues of Vt, which repositions in response to force application in the $F_{pointed}$ direction (Fig. 2b). Specifically, the side chain carboxyl of E1015 (Vt) forms an H-bond interaction with the side chain amino group of R335 (protomer-P) and the side chain carboxyl group of E1021 (Vt) forms an H-bond interaction with the side chain amino group of R147 (protomer-P) (Fig. 2b). The presence of non-native H-bonds mediated by Vt residues E1015 and E1021 was only observed in the $F_{pointed}$ direction. The directional dependence, high occupancy, and significant force-strengthening characteristics indicate that these DAFS residues play a key role in the formation of directional catch bonds.

### Creation of a suite of DAFS Vt variants
To optimally identify substitutions in Vt that perturb $F_{pointed}$-specific DAFS interactions yet retain Vt structure, we performed in silico mutagenesis of E1015 and E1021 using the Eris molecular modeling suite[34]. Eris employs fast sidechain packing and backbone relaxation algorithms to quantify changes in protein structural stability upon mutagenesis[34]. In silico computational mutagenesis using the Eris molecular modeling suite predicted that alanine substitutions at E1015 and E1021 as well as the E1015-E1021 double variant are stabilizing ($\Delta\Delta G < 0$) and retain Vt structure and Vt: F-actin interactions in the unloaded state.

To assess the ability of these variants to prevent DAFS interactions between Vt and F-actin in the $F_{pointed}$ direction, we introduced alanine substitutions at E1015 and E1021 and performed constant-force pulling DMD simulations. Upon loading in the $F_{pointed}$ direction, the E1015A-E1021A Vt variant exhibited a loss of the interface contact area, salt bridges, and vdW interactions (Supplementary Fig. 5a–c). Similarly, the occupancy and force-strengthening of H-bonds were both drastically reduced (Supplementary Fig. 5d). Overall, these data predict that the E1015A–E1021A variant will ablate directional catch bonding without altering the structure of Vt. Moreover, these analyses predict that engagement of E1015 and E1021 with specific residues on actin protomers (E1015 with R335 on protomer-P; E1021 with R147 on protomer-P) creates a unique binding interface that enhances the stability of the Vt:F-actin complex in the $F_{pointed}$ direction by increasing the number of non-bonded interactions and expanding the interface contact area between Vt and F-actin. These intermolecular interactions across different actin protomers contribute significantly to the formation and stabilization of directional catch bonding between F-actin and Vt.

### DAFS variants do not perturb Vt structure and stability
To evaluate the contribution of DAFS residues in maintaining the conformation and stability of Vt (Fig. 3a), we performed far-ultraviolet (UV), near-UV, and circular dichroism (CD) thermal melt profiling of DAFS variants containing alanine substitutions. All Vt DAFS variants showed similar far-UV CD spectral profiles, indicating minimal effects on the secondary structure relative to WT Vt (Fig. 3b). We also compared the near-UV spectra of the DAFS variants to evaluate tertiary packing interactions between the N- and C-terminus, as Vt has a distinct near UV pattern between 260 and 300 nm (positive/dip) due to tertiary packing of two tryptophan residues located at W912 in the H1/H2 loop and W1058 in the C-terminus[35] (Fig. 3c). The near-UV

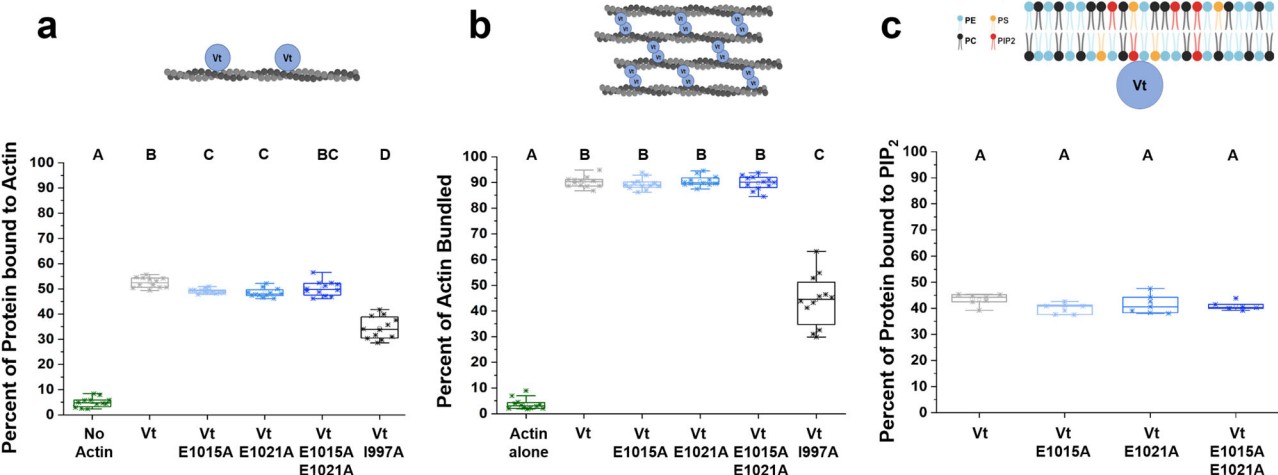

**Fig. 4 | DAFS variants retain Vt interactions with F-actin and PIP₂. a** High speed F-actin co-sedimentation assays comparing actin binding of WT Vt, DAFS and the actin-deficient I997A Vt variant. Vt I997A is an actin-binding deficient control. **b** Low-speed F-actin co-sedimentation assays with WT Vt, DAFS and Vt I997A variants comparing the fraction of F-actin present in bundles or in higher-order assemblies. Box-whisker plots with $n = 12$ independent measurements pooled from $N = 3$ preparations are presented for F-actin **a** binding and **b** bundling co-sedimentation assays, respectively. **c** Box-whisker plot with $n = 7$ independent measurements from $N = 2$ preparations comparing WT Vt and DAFS Vt variant association with PIP2-containing LUVs. Box depicts the median as the center value, the 25th percentile as the lower bound, and the 75th percentile as the upper bound. Whiskers extend 1.5 times the interquartile region (IQR) from the bottom and top of box, or to the minimum and maximum of the data if the data does not extend to the whiskers. Values outside the whiskers are plotted as individual points. One-way analysis of variance (ANOVA) and Tukey's HSD test were used for statistical analysis. Different letters denote significant differences at $p < 0.05$. See Supplementary Tables 1–3 for a detailed listing of p-values. Source data are provided as a Source Data file.

spectral profiles associated with the single (E1015A), (E1021A), and double (E1015A–E1021A) Vt DAFS variants were similar to WT Vt, suggesting that the DAFS variants have similar tertiary structures (Fig. 3c). Moreover, the similarity in the CD melt curves relative to WT Vt indicates that the DAFS variants do not alter thermal stability (Fig. 3d). These data reveal that the single and double DAFS variants do not exhibit altered conformation or stability of Vt.

**DAFS variants retain Vt-F-actin binding and bundling**

To probe the effects of DAFS variants on Vt:F-actin binding, we conducted high-speed actin binding co-sedimentation assays using WT Vt as a positive control and Vt I997A as an actin-binding deficient variant (Fig. 4). We found that the double (E1015A–E1021A) DAFS Vt variant exhibits similar binding behavior to WT Vt when interacting with F-actin (Fig. 4a), and a small reduction (2%) was observed for the E1051A and E1021A variants in comparison to the large 20% reduction observed for Vt I997A. As the binding of F-actin promotes Vt dimerization and F-actin crosslinking, we also quantified the F-actin bundling of the Vt DAFS variants using low-speed actin co-sedimentation assays. We found that single (E1015A, E1021A) and double (E1015A-E1021A) Vt variants can bundle actin similar to WT Vt (Fig. 4b). While E1015A and E1021A DAFS variants exhibit slightly lower (~2%) F-actin binding relative to WT Vt, the actin-bundling ability of both single variants remained unperturbed. This suggests that the slight reduction in actin binding is functionally insignificant. Overall, our biochemical data showed that the single and double DAFS Vt variants did not perturb interactions with F-actin.

**DAFS variants retain Vt:PIP₂/lipid interactions**

Vcn specifically associates with the acidic phospholipid, phosphatidylinositol-4,5-bisphosphate (PIP₂), to drive association with the membrane[28,36]. To determine whether the DAFS variants affect Vt:lipid interactions, we performed lipid co-sedimentation studies with 100 nm large unilamellar vesicles (LUVs) containing phosphocholine (PC), phosphoethanolamine (PE), and 20% phospho-L-serine (PS) and 10% PIP₂[28,35] (Fig. 4c and Supplementary Fig. 6). These results indicate that Vt single and double DAFS variants retain PIP₂ specificity and

exhibit strong association with anionic PIP₂ containing LUVs similar to WT Vt.

**DAFS variants do not perturb FA morphometrics**

Next, we sought to evaluate the effects of DAFS variants on Vcn *in cellulo*. Constructs encoding single and double DAFS Vcn variants tagged with the FP mVenus (VcnVenus) were created and transiently expressed in Vcn⁻/⁻ mouse embryonic fibroblasts (MEFs). To determine the effects on FA initiation, FA maturation, FA elongation, and overall FA organization, the number of FAs per cell, FA size, FA major axis to minor axis ratio, and standard deviation in FA orientation were quantified. As shown in Fig. 5, FA morphometrics quantified at matched expression levels (determined by local Venus intensity) did not change with the introduction of the DAFS Vcn variants. This is consistent with previous work demonstrating that FA morphometrics remain similar across cells expressing WT Vcn or F-actin binding deficient Vcn I997A[30].

**DAFS variants do not affect Vcn Activation**

Vcn is subject to classical head-tail inhibition[37]. Dysregulation of the interface that mediates this interaction can lead to activation of Vcn and excessive accumulation of Vcn within FAs[38]. A FRET-based biosensor incorporating one FP in the strap region of Vcn and another at the C-terminus reports changes in Vcn conformation associated with activation, both within FAs and the cytoplasm[25,26]. To determine whether DAFS Vcn variants affect Vcn activation, Vcn conformation sensors (VcnCSs) harboring no, one, or both variants were created and transiently expressed in Vcn⁻/⁻ MEFs. All VcnCSs localized to FAs as expected, and FA morphometrics for all VcnCSs matched those observed in the VcnVenus constructs (Supplementary Fig. 7), indicating that the presence of the FP within the Vcn strap region did not affect the function of VcnCS or the VcnCS variants. Furthermore, FRET efficiencies within the cytosol were similar for all VcnCSs and consistent with estimates of closed VcnCS obtained in cells plated on poly-L-lysine to prevent FA formation and Vcn activation (Supplementary Figs. 8, 9). At FAs, FRET efficiencies for VcnCS and all DAFS VcnCS variants were not different and consistent with the presence of active

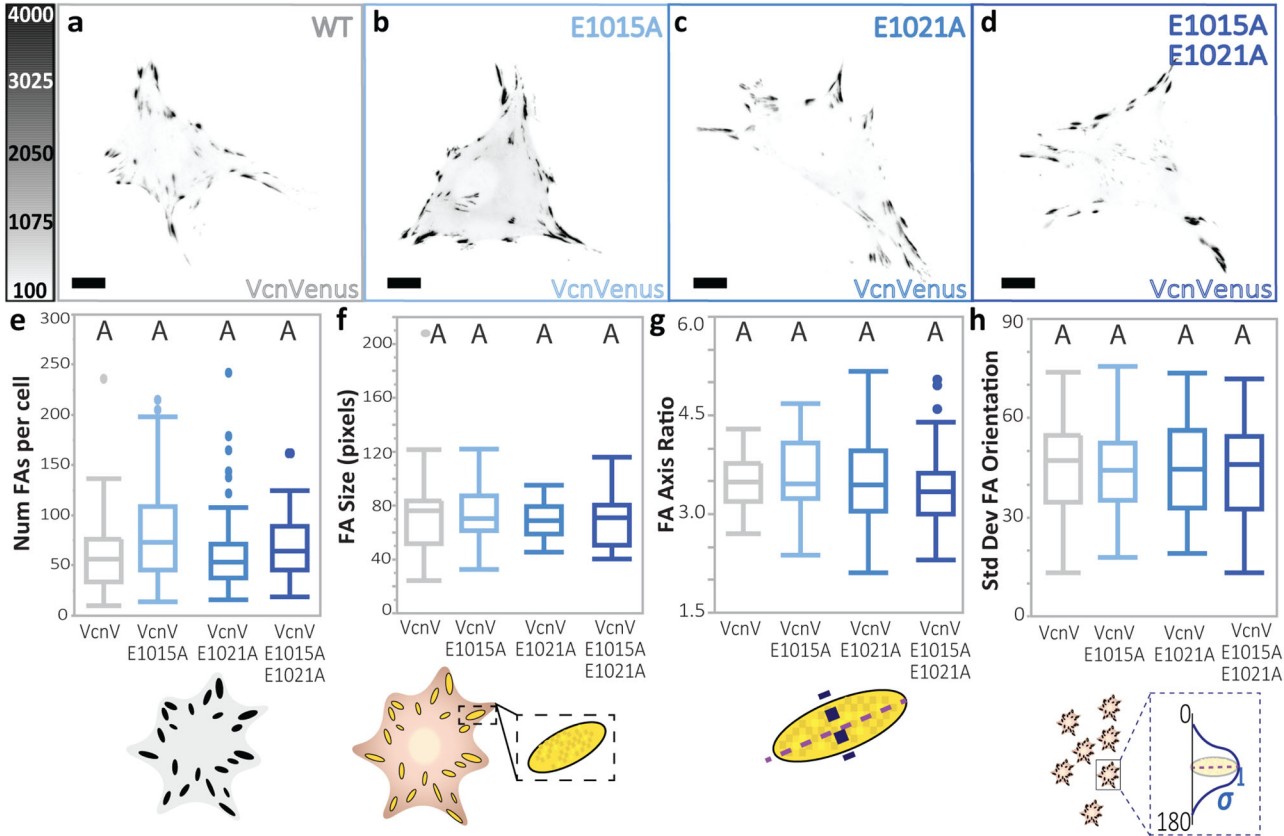

**Fig. 5 | Vcn DAFS variants do not display altered FA characteristics.**
**a**–**d** Representative images of Vcn$^{-/-}$ MEFs expressing WT VcnVenus or VcnVenus containing DAFS variants. Scale bar: 10 µm. WT VcnVenus is shown in gray and VcnVenus DAFS variants are shown in shades of blue. Quantification of FA morphometric characteristics are shown including **e**, FA number, **f**, FA area, **g** FA axis ratio, and **h**, standard deviation of FA orientation. Plots shown for WT VcnVenus and VcnVenus DAFS variants ($n$ = 45, 52, 59, 51 cells, respectively, collected over $N$ = 3 independent experiments). Box depicts the median as the center value, the 25$^{th}$ percentile as the lower bound, and the 75$^{th}$ percentile as the upper bound.

Whiskers extend 1.5 times the IQR from the bottom and top of box, or to the minimum and maximum of the data if the data does not extend to the whiskers. Values outside the whiskers are plotted as individual points. One-way ANOVA paired with a Steel-Dwass test (**e, f, g**) or a Tukey's HSD test (**h**) was conducted across all VcnVenus, VcnCS, and VcnTS constructs used for statistical analyses. Differing letters denote significant difference at $p < 0.05$. For more than two comparisons, see Supplementary Tables 4–7 for a detailed listing of p-values. Source data are provided as a Source Data file.

Vcn (Supplementary Fig. 10). Together, these data demonstrate that the DAFS variants do not affect Vcn conformation.

### DAFS variants progressively, but incompletely, unload Vcn
The engagement of a catch bond is thought to enable the support of larger mechanical loads[11]. Therefore, we sought to determine the effects of DAFS variants on the loads supported by Vcn. A FRET-based biosensor that harbors a tension sensing module (TSMod), composed of two FPs and an extensible domain, within the Vcn strap region has previously been used to report mechanical loading of Vcn in diverse contexts[26,39,40] (Fig. 6a, b). Previous work has established that unloaded TSMod exhibits a FRET efficiency of 28.6% and that incorporation of TSMod into Vcn perturbs the function of neither[26,39–42]. To determine whether DAFS variants affect the loads experienced by Vcn, Vcn tension sensors (VcnTSs) harboring none, one or both DAFS variants were created and transiently expressed in Vcn$^{-/-}$ MEFs. All VcnTSs localized to FAs as expected, and FA morphometrics for all VcnTSs matched those observed in the VcnVenus and VcnCS constructs (Supplementary Fig. 11), indicating that the presence of the TSMod within the Vcn strap region did not affect the function of VcnTS or the VcnTS variants. To account for sensor localization differences between the FAs and the cytoplasm, an actin-binding deficient VcnTS I997A[39,41] and a cytosol-expressed TSMod[26,39,42] (Fig. 6a, top) that lacks flanking Vcn domains were used as unloaded controls. All VcnTSs localized to FAs as expected, and FA morphometrics for all VcnTSs matched those

observed in VcnVenus constructs (Supplementary Fig. 12), indicating that the incorporation of the TSMod did not affect the function of Vcn or the Vcn variants. As we have done previously[39], we used standard segmentation algorithms to study VcnTSs within FAs and the cytoplasm separately (Supplementary Fig. 12a–i). Within the cytoplasm, FRET efficiencies for all VcnTSs are similar to measurements of TSMod and consistent with estimates of unloaded Vcn (Fig. 6 and Supplementary Fig. 12j). Within FAs, VcnTS I997A also appears unloaded. As we have noted previously, the agreement of these various controls indicates that there are no effects due to limited rotation of the FPs in these sensors[26,39,40]. VcnTS supports significant tension (~2.5 pN), indicated by the reduced FRET efficiency (~20%), as expected. Remarkably, with the successive inclusion of DAFS variants, FRET efficiency increases, indicating that Vcn load was reduced at FAs, with a minimal load of 1.3 pN for VcnTS E1015A-E1021A (Fig. 6c–o). The support of an intermediate level of loading is expected for loss of catch bonding but not loss of Vcn:F-actin interactions. Thus, these data show that DAFS residues contribute to the ability of Vcn to bear load and are consistent with DAFS residues mediating Vcn catch bonding.

### DAFS variants have perturbed dynamics and altered tension-sensitivity
The hallmark of a catch bond is an increase in binding lifetime with increased mechanical loading[14]. To probe the relationship between protein exchange dynamics amongst subcellular structures and

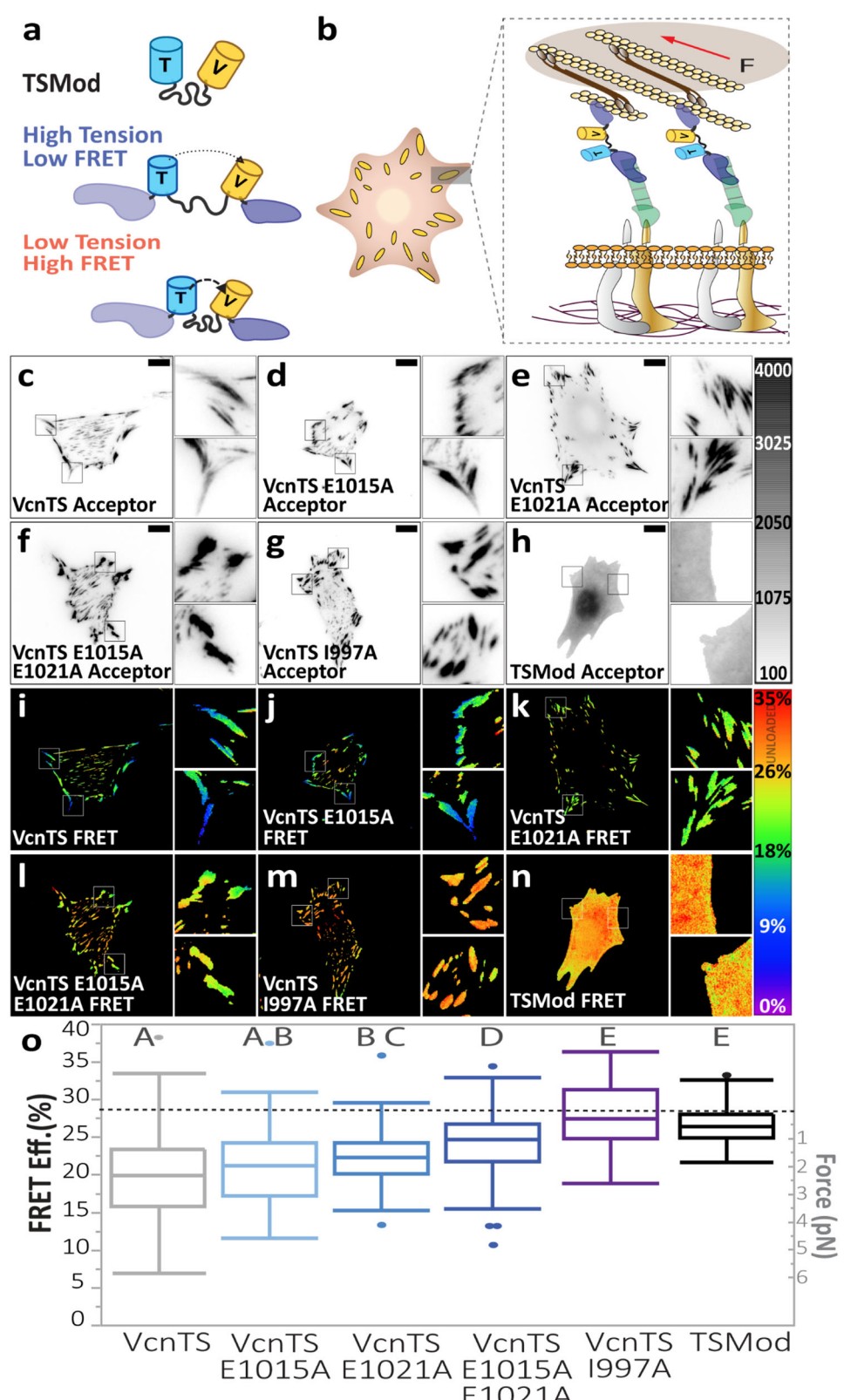

molecular tension, we previously combined imaging of FRET-based molecular tension sensors with sensitized emission and Fluorescence Recovery After Photobleaching (FRAP), establishing the FRET-FRAP technique[39,43]. To probe the effect of DAFS variants on the tension-sensitive exchange dynamics of Vcn, we performed FRET-FRAP on Vcn[−/−] MEFs stably expressing VcnTS and the DAFS variant that exhibited the largest change in loading, VcnTS E1015A-E1021A, at

similar levels (Supplementary Fig. 13a). To ensure the comparability of the selected FAs from these cell types, we verified that the cell area, FA area, and FA acceptor intensity were all matched between the samples, while the differences in FRET between the constructs were maintained (Supplementary Fig. 13b–g). After photobleaching, quicker recovery of acceptor intensity, indicating faster exchange dynamics, was observed in the VcnTS harboring DAFS variants (Fig. 7a–c). To quantify the

**Fig. 6 | DAFS variant VcnTSs experience reduced, but not negligible load.**
**a** Schematic of TSMod and VcnTS constructs in loaded and unloaded states.
**b** Schematic of VcnTS in the FA, loaded by actomyosin contractility. **c–g** FA masked acceptor intensity images are shown for single Vcn$^{-/-}$ MEFs expressing WT VcnTS, DAFS variants of VcnTS and a previously validated variant with reduced actin affinity, VcnTS I997A. **h** Representative cytoplasm acceptor intensity shown for a single Vcn$^{-/-}$ MEF expressing unloaded FRET control, TSMod. **i–m**, Corresponding FA masked FRET efficiency shown for (**c–g**). **n** Corresponding cytoplasm FRET efficiency shown in **h**. **o** Box-whisker plots are shown for cell-averaged FRET efficiency of VcnTS and VcnTS DAFS variants compared to expected unloaded FRET efficiency (dotted line) controls ($n = 232, 85, 79, 97, 58,$ and 69) biologically

independent cells respectively, examined from over $N = 3$ independent experimental days. Box depicts the median as the center value, the 25th percentile as the lower bound, and the 75th percentile as the upper bound. Whiskers extend 1.5 times the IQR from the bottom and top of box, or to the minimum and maximum of the data if the data does not extend to the whiskers. Values outside the whiskers are plotted as individual points. One-way ANOVA paired with a Steel-Dwass test was used for statistical analysis. Differing letters denote significant difference at $p < 0.05$. See Supplementary Table 8 for a detailed listing of exact p-values. Source data are provided as a Source Data file. Scale bars are 10 µm in all images, and insets are 10 µm in length and width.

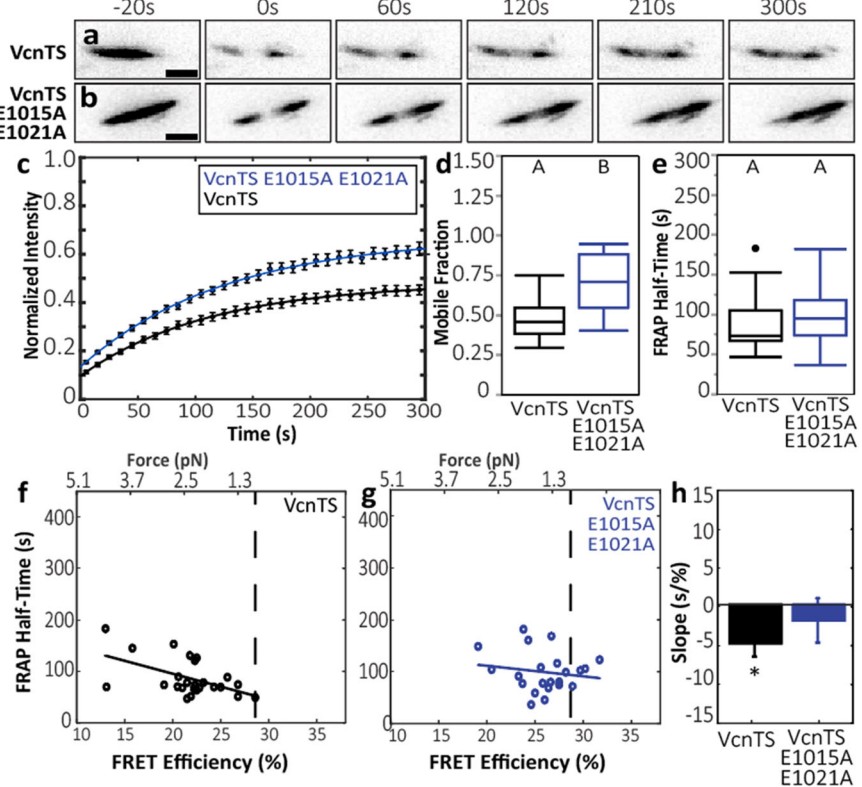

**Fig. 7 | DAFS variant VcnTSs exhibit loss of Vcn force-stabilized dynamics.**
**a, b** Representative Vcn$^{-/-}$ VcnTS and Vcn$^{-/-}$ VcnTS E1015A-E1021A FAs undergoing FRAP time lapse. Scale bar: 2 µm **c** FRAP recovery curves are shown for cells expressing VcnTS and VcnTS E1015A-E1021A ($n = 25, 22$ from $N = 11$ independent experimental days). Measure of center for error bars is average of n normalized intensities at respective timepoint for VcnTS or VcnTS E1015A-E1021A expressing cells. Exponential fit curves are fitted to averages at respective timepoints. **d** Box-whisker plots of mobile fraction of FAs analyzed in c, using two-tailed Welch's t-test for unequal variances (p = 0.0000054) **e** Box whisker plots of FRAP recovery half-time of FAs analyzed in **c,** using one-way ANOVA followed by F-test ($p = 0.25$). Box depicts the median as the center value, the 25th percentile as the lower bound, and the 75th percentile as the upper bound. Whiskers extend 1.5 times the IQR from the

bottom and top of box, or to the minimum and maximum of the data if the data does not extend to the whiskers. Values outside the whiskers are plotted as individual points. **f** Correlation between FRAP recovery half-time and FRET efficiency for VcnTS corresponds to the force-stabilized state. **g** Correlation between recovery half-time and FRET efficiency for VcnTS E1015A-E1021A corresponds to the force-independent state. **h** Regression slopes calculated for panel (**f–g**). A least-squares linear regression was fit to the FRAP half-time and FRET efficiency data for each construct. Bars represent linear regression slope, and error bar represents standard error of the regression slope. Star indicates slope is statistically significant from zero as detected by one-way ANOVA followed by an F-test ($p = 0.007$ for VcnTS; $p = 0.49$ for VcnTS E1015A-E1021A). Source data are provided as a Source Data file.

differences in Vcn exchange dynamics, we fit the FRAP data to a standard exponential curve defined by two parameters: a half-time of recovery and a mobile fraction[39,44]. Estimates of Vcn mobile fraction and half-time are similar to previous measurements[26,38,39,45]. Vcn DAFS variants increased the mobile fraction, demonstrating an increase in the amount of Vcn able to exchange in the experimental window, but a statistically significant change in exchange dynamics was not observed (Fig. 7d, e). Consistent with reports of Vcn catch bonding, we previously demonstrated that Vcn exhibits tension-stabilized exchange dynamics, in which half-time increases with increasing tension (decreasing FRET)[39]. Notably, relationships were not always observable from the analysis of recovery half-time alone. Therefore, we

investigated the relationship between half-time and molecular tension (Fig. 7f, g). For VcnTS, we observed a positive correlation, indicative of tension-stabilized exchange dynamics in Vcn (Fig. 7f), as we previously observed. For the DAFS variant VcnTS E1015A–E1021A, no correlation was observed, indicating a loss of tension-stabilized exchange dynamics. No correlations between half-time and FRET efficiency, half-time and initial acceptor intensity, or FRET efficiency and initial acceptor intensity were observed (Supplementary Fig. 14). Altogether, these data indicate that the inclusion of DAFS variants E1015A and E1021A in Vcn prevents tension-stabilized exchange dynamics, consistent with a loss of Vcn:F-actin catch bonding *in cellulo*.

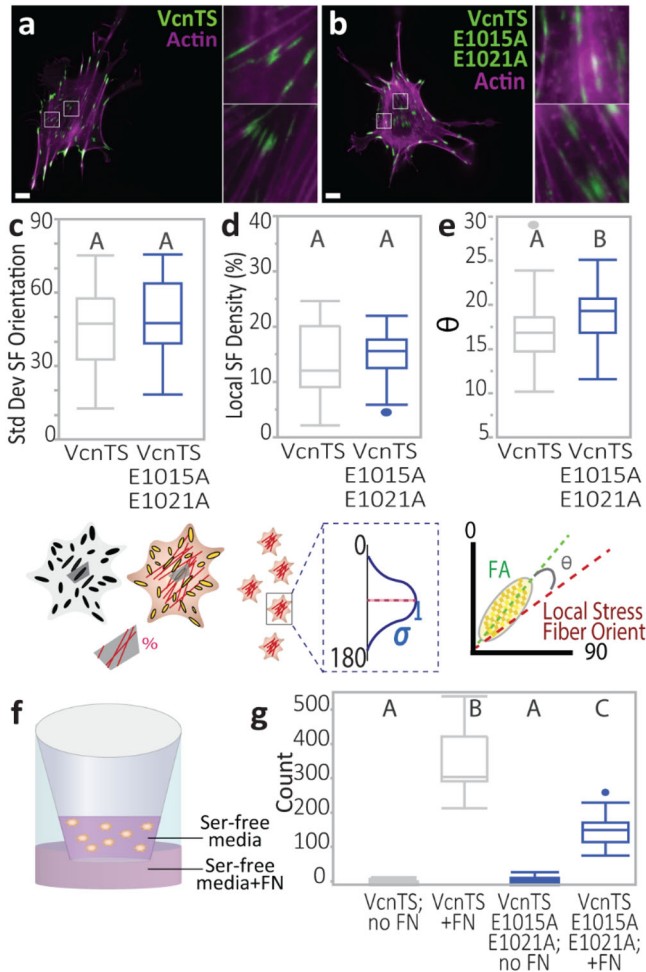

**Fig. 8 | Local FA-SF alignment and directed migration ability differ between WT VcnTS and DAFS double variant VcnTS. a** VcnTS or **b**, DAFS double variant VcnTS E1015A-E1021A stained for F-actin (purple). Scale bar (and zoom-in height and width): 10 μm. Box-whisker plots shown for stably expressed VcnTS or VcnTS E1015A-E1021A in Vcn$^{-/-}$ MEFs (n = 36, 45 respectively, from N = 3 independent experimental days) quantifying **c** spread of actin SF orientations (p = 0.34, one-way ANOVA followed by F-test) **d** SF density per FA-defined local region (p = 0.45, two-tailed Student's t-test paired with post-hoc Tukey Test), and **e** angle between FA and average angle of actin filaments within respective Voronoi regions of cells (p = 0.0173, one-way ANOVA followed by F-test). **f** Schematic depicting transwell setup for directed migration assay. **g** Box-whisker plots are shown of migrated cells per field of view (FOV) in a Boyden chamber haptotaxis migration assay (n = 19,18,16,18 FOVs per given group from N = 3 independent experiments). Box depicts the median as the center value, the 25th percentile as the lower bound, and the 75th percentile as the upper bound. Whiskers extend 1.5 times the IQR from the bottom and top of box, or to the minimum and maximum of the data if the data does not extend to the whiskers. Values outside the whiskers are plotted as individual points. See Supplementary Table 9 for a detailed listing of exact p values of 8 g. Source data are provided as a Source Data file.

## DAFS residues mediate co-alignment of SFs and FAs

While simulations have suggested the involvement of directional catch bonds between Vcn and F-actin in mediating the coordination of FAs and the actin cytoskeleton[11], experimental data to support this hypothesis are currently lacking. FAs undergo a maturation, or strengthening, process that is associated with a variety of distinct F-actin-based structures that have differential compositions, alignments, and orientations[46,47]. Specifically, immature FAs are associated with protrusive (PR) structures comprised of flowing, less aligned F-actin[48]. Mature FAs are associated with stress fibers (SF), which contain highly

aligned F-actin[49]. Therefore, we first sought to determine if incorporation of DAFS variants differentially affects Vcn in either of these FA classifications. As the functions of Vcn and VcnTS have been shown to be nearly identical, cells expressing VcnTSs were used for these experiments, as we have done before[26,41]. To do so, we again focused on Vcn$^{-/-}$ MEFs stably expressing either WT VcnTS or VcnTS E1015A−E1021A. As incorporation of the double variant in VcnTS did not affect FA morphometrics (Fig. 5), we used a previously established morphometrics-based machine learning classification algorithm that is compatible with the imaging of FRET-based biosensors by sensitized emission to identify PR-associated, immature FAs and SF-associated, mature FAs (Supplementary Fig. 15a, b)[41]. Previous analyses of the VcnTS-expressing cells revealed that average loads within the two FA classifications were similar, but that SF-associated FAs exhibited a spatial gradient in loading, with the largest loads being membrane proximal[41]. Manual quantification of subsets of identified FAs with line scans, as well as automated analyses of the entire data set, revealed that the previously established relationships are maintained in this analysis of VcnTS (Supplementary Fig. 15). VcnTS E1015A-E1021A exhibited an equivalent loss of loading in both types of FAs, consistent with the whole cell analyses above (Fig. 6). Interestingly, a loss of the spatial gradients in loading observed in SF-associated FAs of VcnTS E1015A-E1021A (Supplementary Fig. 15p) suggests a more prominent role for directional catch bonding in mature, stress fiber-associated FAs.

Next, we sought to probe the coordination of FAs and SFs. Vcn$^{-/-}$ MEFs stably expressing either VcnTS or VcnTS E1015A-E1021A were allowed to spread for 24 hours to favor mature FA and SF formation (Fig. 8a, b). The standard deviation of SF orientation was quantified to assess overall F-actin organization, and no differences were found between VcnTS and VcnTS E1015A-E1021A expressing cells (Fig. 8c). To probe the coordination of FAs and the actomyosin cytoskeleton, Voronoi tessellation was used to define local FA-centered regions of SFs within the cells. Local SF density was calculated within each region, and no significant differences were observed (Fig. 8d), suggesting that there were no variations in F-actin assembly. Next, vectors corresponding to the local orientation of the FA and the local orientation of the SFs were generated (Supplementary Fig. 16). The angle between these vectors was taken as a measure of the coordination between FAs and stress fibers, with a smaller difference indicative of increased coordination. VcnTS E1015A-E1021A expressing cells showed reduced co-alignment compared to VcnTS-expressing cells (Fig. 8e), consistent with a key role for DAFS residues in mediating coordination of FAs and the actomyosin cytoskeleton. Overall, these data are consistent with a key role for DAFS residues in mediating the coordination of mature FAs and stress fibers.

## DAFS residues are required for optimal haptotaxis

Vcn is a critical mediator of directional migration[50,51]. To determine if DAFS residues are important in directed cell migration, Vcn$^{-/-}$ MEFs expressing either WT VcnTS or VcnTS E1015A-E1021A were subjected to a Boyden chamber−based haptotaxis assay (Fig. 8f and Supplementary Fig. 17). Nearly double the number of VcnTS MEFs underwent haptotaxis, compared to the VcnTS E1015A-E1021A MEFs (Fig. 8g). Thus, DAFS residues, and likely the engagement of the Vcn:F-actin catch bond, promote directed cell migration[11].

## Discussion

The ability of mammalian cells and tissues to resist or deform differentially in response to mechanical forces applied in distinct directions is mediated by the ability of load-bearing proteins to maintain or change connectivity in response to these forces[52,53]. In the context of cell-ECM adhesion, previous work has identified several key examples, and simulations have predicted key roles for directional catch bonds in polar or directed molecular and cellular process[11,17]. To test these

predictions, we developed an integrated procedure enabling the in silico identification, rational engineering, biochemical validation, and probing of the *in cellulo* roles of amino acids mediating directional catch bonding. Specifically, we employed a combination of DMD simulations, in silico mutagenesis, controlled biochemical assays, FRET-based biosensors, photobleaching, quantitative image analysis, and cell migration assays to fully characterize amino acid variants that perturb directional catch bonding. Our findings indicate that the Vcn DAFS variants have minimal impact on other cellular functions, allowing us to specifically evaluate alterations in cell substructures and overall cellular behavior.

We focused on the mechanical linker protein Vcn, as it mediates force transmission in multiple subcellular structures and exhibits the strongest directionally sensitive catch bond described to date[11]. Our initial analysis of the umbrella sampling simulations provided valuable insights into the energetics and stability of the Vt:F-actin complex under different pulling directions and provided a key connection to experimental observations of directional catch bonding[11]. It is worth mentioning here that while umbrella sampling is a valuable technique for exploring the energetic aspects of molecular interactions under force, it may not accurately quantify binding affinity[54]. However, the approach is justified here as the relevant proteins (F-actin and Vt) are relatively stiff, the forces are low, and the conformation changes are small. These criteria may not be met in systems with soft biopolymers (i.e. single stranded DNA[12,55]) where multiple conformers are expected or in scenarios involving large force and substantial protein conformational changes (i.e. bacterial adhesion proteins[33,56,57]). Furthermore, we have purposefully only made qualitative comparisons (e.g., the ordering of the bond strength as function pulling direction), as interpreting the simulation results quantitatively (e.g. as an indication of bond affinity) may represent an over-interpretation of the data. Several alternative methodologies, such as steered molecular dynamics (SMD)[58], enhanced sampling techniques such as metadynamics[59] or replica exchange simulations[60], may be required for such a detailed investigation of catch bonding energetics between Vt and F-actin.

The subsequent DMD simulations of the Vt:F-actin complex in different directions relative to F-actin captured force-induced conformational changes and non-covalent interactions that contribute to catch bonding. H-bond occupancy analyses suggested that the directional catch bonding between Vt and F-actin was driven by DAFS H-bond formation by E1015 and E1021. In silico mutagenesis studies identified E1015A, E1021A, and E1015A-E1021A as single and double Vt variants that should ablate pertinent interactions without perturbing Vt structure. DMD simulations using the Vt E1015A-E1021A double variant revealed a lack of formation of key DAFS H-bonds with F-actin. Moreover, this variant exhibited an enhanced disruption of various other intermolecular interactions when subjected to forces in the $F_{pointed}$ direction, establishing a key role of H-bonds formed by E1015 and E1021 in reinforcing a variety of other types of intermolecular interactions to facilitate strong catch bonding (Supplementary Table 21). Altogether, these findings strongly indicate that residues E1015 and E1021 play a pivotal role as key mediators of Vcn's directional catch bonding.

To experimentally probe the function of DAFS residues, a broad survey of their effects on Vcn behavior was performed. For in vitro analyses, Vt domains harboring DAFS variants (E1015A, E1021A, and E1015A–E1021A) were evaluated. Structural analysis using near, far, and temperature-dependent CD measurements revealed that DAFS variants do not alter Vt structure. Actin co-sedimentation assays showed that the DAFS Vt variants maintained the ability to bind and bundle F-actin. Lipid co-sedimentation revealed that the binding strength and binding specificity to phospholipids is not affected by Vt DAFS variants. These data show that DAFS variants do not perturb Vt structure or some key Vcn functions. *In cellulo* evaluations of DAFS variants in Vcn-Venus revealed no effects of FA morphometrics, which is broadly

consistent with previous work involving weakening of Vcn:F-actin interactions[61]. The incorporation of DAFS variants into VcnCSs revealed no alterations in Vcn activation[26], consistent with previous reports of the independent regulation of Vcn activation and loading. Interestingly, the incorporation of DAFS variants into VcnTSs revealed an incomplete loss of loading, as expected for loss of catch bonding. Analysis of the VcnTS exchange dynamics between FAs and the cytosol with FRAP revealed that the incorporation of the double DAFS variant increased the fraction of Vcn undergoing exchange. Furthermore, analysis of the correlation between the tension experienced by Vcn and its exchange dynamics revealed tension-stabilization, as we previously reported for Vcn and is expected for a catch bond[39]. This correlation was lost in the double DAFS variant, indicating a loss of tension-stabilized exchange dynamics and consistent with the loss of Vcn catch bonding. Overall, these data are consistent with the DAFS variants perturbing catch bonding between Vcn and F-actin *in cellulo*, without perturbing other Vcn functions.

Next, we probed the effects of the loss of Vcn directional catch bonding on adhesion-associated, mechanosensitive subcellular, and cellular processes. To begin, we determined that DAFS variants affect Vcn more in mature, SF-associated FAs than in immature, PR-associated FAs. This is consistent with predictions of key roles for Vcn directional catch bonding in structures with aligned F-actin[11]. Analyses of SF formation revealed no defects due to the expression of the double DAFS Vcn variant. However, cells expressing this double DAFS variant Vcn showed defects in the coalignment of FAs and SFs, consistent with the prediction that directional catch bonds are important in mediating the coalignment of these structures[11,17]. To probe a cellular process, we focused on directed migration in response to a haptotactic gradient. We found that cells expressing the double DAFS Vcn variant exhibited substantially reduced directed migration, which is broadly consistent with our previous work demonstrating the importance of Vcn:F-actin interactions in directed cell migration[39]. This result establishes a key role for Vcn catch bonding in a fundamentally important cellular process. Overall, these data are consistent with a model in which other mechanisms dictate the assembly of FAs and SFs, but Vcn directional catch bonding mediates the coordination of these subcellular structures to enable complex cellular behaviors, such as directed migration[62]. Interestingly, a recently developed mathematical model suggests that the self-stabilization of linker proteins (e.g., Vcn binding to unfolded talin domains and preventing refolding and not the engagement of catch bonds) is the dominant driver of adhesion maturation[63]. A key issue for future work is further elucidating the roles of self-stabilization and directional catch bonding in mediating the coordinated assembly and coordination of FAs and stress fibers, particularly in the context of cell migration.

Based on our findings, we propose a working model of the mechanism of directional catch bonding between Vcn and F-actin, as illustrated in Supplementary Fig. 19. Overall, catch bonding seems to be due to a combination of force-induced structural changes in Vt and the asymmetric nature of F-actin. Initially, when the pulling force is applied in the $F_{pointed}$ direction, Vt strongly adheres to protomer-B of F-actin. This causes a reorientation of Vt along the short axis of F-actin as well as rearrangement of the helical bundles within Vt. These changes bring Vt closer to F-actin and expose the side chains of E1015 and E1021, enabling the formation of non-native H-bonds with R335 on protomer-P and R147 on protomer-P, respectively (Supplementary Fig. 18). The H-bonds between E1015/E1021 in Vt and R335/R147 in F-actin create a unique binding interface that has an enhanced contact area and reinforces a variety of intermolecular interactions that enable the substantial strength on the Vcn catch bond when forces are applied in $F_{pointed}$ direction. Notably, the twisting of Vt, rearrangement of helical bundles, and exposure of E1015 and E1021 are not observed when Vt is pulled in the $F_{barbed}$ direction. Therefore, forces in the $F_{barbed}$ direction result in the formation of distinctly weaker

intermolecular interactions, as existing H-bonds are strengthened to a lesser degree and salt-bridges and hydrophobic interactions are not reinforced. This directional asymmetry is likely due to the structural heterogeneity and intrinsic polarity of F-actin[64] (Supplementary Fig. 19). An important question for future work will be determining whether the reinforcement observed in other types of intermolecular interactions by forced-induced H-bond formation is a common scenario in directional catch bonding.

To begin to assess the generalizability of this model, we compared Vcn and α-catenin (Catn), which are structural homologs recruited to AJs in response to applied loads and exhibit directional catch bonding with F-actin[65,66]. Cryo-EM reconstructions of F-actin bound Catn (PDB ID: 6UPV [https://doi.org/10.2210/pdb6upv/pdb] (αE-catenin ABD-F-actin complex))[65] and Vt (PDB ID: 3JBI [https://doi.org/10.2210/pdb3jbi/pdb] (vinculin tail domain bound to F-actin))[29] show similar engagement to F-actin (RMSD = 0.99 Å,) and demonstrated that key catch bond residues in Vt have a comparable orientation to Catn at the actin interface (Supplementary Fig. 20). Specifically, M816 of Catn, which is predicted to form force-sensitive catch bond interactions with F-actin[66], is positioned and oriented similarly to E1021 in Vt, despite having different biochemical properties as an uncharged and hydrophobic residue. Another noteworthy comparison is between D813 of Catn and E1015 of Vcn. Although both residues are charged, they are positioned and oriented differently in the two structures (Supplementary Fig. 20). We postulate that these differences are what lead to the similar, but weaker, directional catch bonding observed in Catn:F-actin (Supplementary Fig. 1c).

Overall, this work develops and implements an interdisciplinary approach for identifying, perturbing, and assessing key biological functions of directional catch bonding between F-actin and Vcn. In this work we focused on cell–ECM adhesion and single cell migration. Vcn also has key roles in cell–cell adhesion, collective cell migration, and a variety of developmental as well as pathophysiological mechanosensitive processes where directional catch bonding is likely to play key, but undefined, roles. These can now be explored with the tools and approaches developed here. Furthermore, this approach will likely be generalizable to other F-actin binding, loading-bearing proteins and will enable the assessment and potential manipulation of directional catch bonding in this large class of proteins. and mediate diverse processes, such as binding of neutrophils to inflamed endothelial cells[67] and T-cell receptor activation[68]. Directional catch bonds also play a key role in bacterial adhesion, but the relevant forces and protein conformations are much larger and more substantial than those involved in mammalian cell adhesion[33,56,57]. An important question for future work will be determining if similar or distinct mechanisms mediate directional catch bonding in these mechanically distinct regimes. Overall, we expect that the integrated procedure for identifying key residues and producing tools to probe the effects of the loss of directional catch bonding developed here will facilitate a plethora of studies in mechanobiology at the molecular, cellular, and tissue level across these wide-ranging fields.

## Methods

### Umbrella sampling simulations to quantify the interaction energy between F-actin and Vt

To generate a series of F-actin–Vt structural conformations with increased center-of-mass (CoM) distance between F-actin and Vt, we considered the cryo-EM structure F-actin:Vt[30] and pulled Vt away from the actin filament along $F_{pointed}$, $F_{barbed}$, and $F_{normal}$ directions separately. During the pulling simulations, position restraints were applied on backbone heavy atoms of F-actin to keep it immobile and no position restraints were applied on Vt. In each direction, the pulling simulations were executed for 1000 ps, using a pull rate of 0.01 nm/ps and a spring constant of 1000 kJ/mol/nm². From each pulling

trajectory, snapshots of F-actin–Vt were extracted and utilized as starting configurations for the umbrella sampling windows[69]. Each of these configurations corresponds to harmonically restrained Vt away from F-actin, via an umbrella biasing potential. The harmonic restraint between F-actin and Vt allows Vt to sufficiently explore the configurational space in each window along a reaction coordinate ($\xi$)[69]. We chose a window spacing between 1 – 20 Å CoM separation between F-actin and Vt, which resulted in 26 windows in each pulling trajectory. The F-actin–Vt configuration in each window was solvated with water molecules, and 150 mM NaCl was added to neutralize the charge of the system. We chose the CHARMM36 forcefield[70] to generate bonded and non-bonded interaction parameters for all atoms in the system, and performed steepest descents energy minimization followed by position restraints equilibration for 100 ps under constant-temperature, constant-volume (NVT) ensemble. During the equilibration, protein and non-protein atoms were separately coupled to temperature baths and maintained at a temperature of 310 K using the Berendsen weak coupling method[71]. Subsequently, we carried out 100 ps of equilibration in a constant-temperature, constant-pressure (NPT) ensemble to maintain isotropic pressure at 1.0 bar using the Parrinello-Rahman barostat[72]. Finally, we employed Nosé- Hoover thermostat[73,74] and the Parrinello-Rahman barostat[72] for a 10 ns production run in each window with position restraints on F-Actin. We calculated long-range electrostatics using the particle mesh Ewald (PME) algorithm[75] and chose the cut-off of short-range nonbonded interactions at 14 Å. All MD simulations were conducted using the GROMACS-2018 package[76]. Please refer to Supplementary Table 22 for a detailed list of the parameters used in the execution of umbrella sampling molecular dynamics simulations. We derived potential of mean force (PMF) from each of the simulations, assembled the PMF curve with respect to the reaction coordinate, and estimated the binding energy (ΔG) between F-actin and Vt in three pulling directions using the weighted histogram analysis (WHAM) method[77].

### Constant-force pulling discrete molecular dynamics (DMD) simulations to identify force-dependent interactions between actin and Vcn

The 8.50 Å cryo-EM reconstruction of the F-actin/Vt complex[30] was used as a starting structural model for constant-force pulling simulations. The missing N- and C-terminal residues of Vt were added by homology modeling using the high-resolution X-structure of Vt (PDB ID: 1QKR [https://doi.org/10.2210/pdb1qkr/pdb] (vinculin tail))[78] as the template. The best F-actin–Vt complex model with optimized modeler objective function was selected out of forty generated models using Modeller-9v19[79]. The optimized structure was placed in a cubic box with side-length 500 Å with periodic boundary conditions. The Anderson thermostat was used to maintain temperature at 300 K (room temperature). Constant-force pulling was achieved by applying a discretized step-function with a constant energy jump, dE, at the distance step of dR (1 Å) between F-actin and Vt, where the pulling force (f) is estimated as dE/dR, with dE of approximately 0.015 kcal/mol[32]. We selected the center of mass (CoM) of Vt as the reference point for applying external forces. The CoM of Vt was subjected to pulling forces in the $F_{pointed}$, $F_{barbed}$, and $F_{normal}$ directions by employing a discretized step-function with a constant energy jump at equal distance intervals. The position of F-actin was harmonically restrained to maintain its stability during the pulling simulations. Based on previous pulling simulations on biological complexes[32], we chose pulling forces within range 0–150 pN and sampled at an interval of 1 pN. Each pulling simulation at a specific force was carried out for 400,000 DMD steps and repeated 10 times to establish the confidence interval. The data obtained from our simulations was subsequently used to analyze the DAFS H-bonds between F-actin and Vt. Data from simulation trajectories was extracted using in-house scripts and GROMACS analysis tools. The representative 3D conformations of

F-actin and Vt were rendered using PyMOL[80] and visual molecular dynamics (VMD)[81].

To determine H-bond formation between donor (D) and acceptor (A), we employed a cut-off distance of 3.5 Å with the acceptor-donor-hydrogen angle ≤30° [18] and subsequently quantified the occupancy of each H-bond interaction over the entire MD trajectory[82]. H-bond occupancy is defined as a fraction of conformations in which a specific residue pair is involved in H-bond interactions. We converted these occupancy fractions to occupancy percentages by multiplying with '100'.

To visualize the H-bond occupancies and their distribution in the Vt:F-actin complex, we used the Circos tool (version 0.69-8)[83]. Circular plots were generated to represent interactions between Vt and F-actin in different pulling directions.

The computation of salt bridges was performed using the VMD (Visual Molecular Dynamics) software with the built-in Salt Bridges plugin. This plugin identifies and calculates the salt bridges based on the distance criteria between positively and negatively charged residues (d <= 3.2 Å).

To estimate the vdW contacts between Vt and F-actin, the GROMACS tool gmx mindist with distance cut-off of 5 Å was utilized. This tool calculates the minimum distance between pairs of atoms in different molecules, allowing for the identification of vdW contacts and their corresponding distances.

Additionally, the gmx sasa tool from GROMACS was utilized to compute the solvent-accessible surface area (SASA) of the Vt:F-actin interface. This tool calculates the surface area accessible to the solvent molecules, providing insights into the extent of exposure and interface area between Vt and F-actin.

## Computation of fraction of native H-bonds from DMD simulation trajectories

To determine the fraction of native H-bonds, we first identified the native H-bonds present in the initial structure of the complex. These native H-bonds were defined as the H-bonds observed in the experimental or reference structure of the complex. Next, we analyzed the DMD simulation trajectories and counted the number of native H-bonds that were maintained throughout the simulation.

The fraction of native H-bonds ($F_{native}$) in the Vt:F-actin complex was calculated using the following formula:

$$F_{native} = \frac{N_{persistent}}{N_{total}} \quad (1)$$

where $N_{persistent}$ represents the number of native H-bonds that remained intact throughout the simulation, and $N_{total}$ represents the total number of native H-bonds present in the initial structure of the complex (PDB: 3JBI). The fraction of native H-bonds provides a quantitative measure of native H-bonds preserved during the simulation.

## Eris calculations to identify suitable benign DAFS residue substitutions for experimental evaluation

In silico mutagenesis studies on Vt DAFS residues were performed using Eris molecular suite[34]. The high-resolution X-structure of Vt (PDB ID: 1QKR)[78] was used to evaluate residue perturbations. Initially, Eris employs residue substitutions in protein structure and evaluates free energies of native ($\Delta G_{wt}$) and variant ($\Delta G_{var}$) conformations. Then, Eris computes the change in free energy of protein upon amino acid substitution ($\Delta\Delta G_{var}$) by subtracting the free energy of native protein from that of the variant. Based on $\Delta\Delta G_{var}$ values, Eris estimates amino acid substitutions as either stabilizing ($\Delta\Delta G_{var} < 0$) or destabilizing ($\Delta\Delta G_{var} > 0$).

## Plasmid design

For stability and interaction studies, Vt (comprising the chicken sequence 879-1066) was cloned into a pQlinkH vector (Addgene, Cambridge, MA). All DAFS Vt variants, E1015A, E1021A, and E1015A-E1021A were constructed using a Q5 site-directed mutagenesis kit (New England Biolabs, Ipswich, MA). Primers (Eton Biosciences, Durham, NC) used for site-directed mutagenesis on Vt to generate Vt-E1015A include forward primer 5′-CAG CGA TGA AGC CTC AGA ACA GGC AAC-3′ and reverse primer 5′-ATG TTA GTC CTG CCC AGC-3′. To construct Vt-E1021A forward primer 5′-A CAG GCA ACT GCG ATG TTG GTT C-3′ and reverse primer 5′-TCT GAT TCT TCA TCG CTG ATG-3′were used. For cellular studies, E1015A and E1021A variants of VcnVenus and VcnTS were constructed on pcDNA3.1 Vcn Venus (Addgene Plasmid #27300) and the pcDNA3.1 VcnTS plasmid (Addgene Plasmid #26019), respectively. The forward primer 5′-CAG CGA TGA AGC CTC AGA ACA GGC AAC-3′ and reverse primer 5′-ATG TTA GTC CTG CCC AGC-3′as well as forward primer 5′-A CAG GCA ACT GCT ATG TTG GTT C-3′ and reverse primer 5′-TCT GAT TCT TCA TCG CTG -3′were used to generate E1015A and E1021A variants, respectively. Site-directed mutagenesis on the pcDNA3.1 VcnCS plasmid was used to generate variants of VcnCS[25,26]. Primers for site-directed mutagenesis on pcDNA3.1 VcnCS to generate pcDNA3.1 VcnCS E1015A include forward primer 5′-CAG CGA TGA AGC TTC AGA ACA GGC AAC-3′ and reverse primer 5′-ATG TTA GTC CTG CCC AGC-3′. Primers for site-directed mutagenesis on VcnCS to obtain VcnCS E1021A include forward primer 5′-A CAG GCA ACT GCC ATG TTG GTT C-3′ and reverse primer 5′-TCT GAT TCT TCA TCG CTG ATG-3′. To obtain pcDNA3.1 VcnCS E1015A-E1021A, site-directed mutagenesis was performed on the pcDNA3.1 VcnCS E1015A plasmid using forward primer 5′-A CAG GCA ACT GCC ATG TTG GTT C-3′ and reverse primer 5′-TCT GAA GCT TCA TCG CTG-3′. pRRL VcnTS (Addgene Plasmid #111830) and variants of pRRL VcnTS were generated via 5′NruI/3′XbaI digestion and ligation of the Vh-TSMod-Vt sequence from pcDNA 3.1 VcnTS (T4 DNA Ligase; New England BioLabs, Ipswich, MA) into a pRRL vector digested with 5′ EcoRV/3′XbaI. All constructs were verified via DNA sequencing (Azenta Life Sciences, Morrisville NC).

## Protein expression and purification

All pQlinkH expression vectors contain an N-terminal His-tag and TEV cleavage site. WT Vt domain and Vt DAFS variants were expressed in the Escherichia coli BL21 (DE3) strain and purified[84]. Briefly, cells were first grown at 37 °C to an optical density of 0.6–0.8 at 600 nm. Protein expression was then induced by addition of isopropyl-D-1-thiogalactopyranoside (0.5 mM). After induction, cells were grown overnight at 18 °C and harvested by centrifugation at 4400x $g$ for 30 minutes. Cell pellets were resuspended in lysis buffer (20 mM Tris, 150 mM NaCl, 5 mM imidazole, 2 mM β-mercaptoethanol, pH 7.5), lysed by sonication and purified first by affinity separation using Ni-NTA-agarose beads (Qiagen). Vt protein bound to the His-tag beads were exchanged in wash buffer (20 mM Tris, 150 mM NaCl, 60 mM imidazole, 2 mM β-mercaptoethanol, pH 7.5) before eluting in elution buffer (20 mM Tris, 150 mM NaCl, 500 mM imidazole, 2 mM β-mercaptoethanol, pH 7.5). For His-tag removal, the eluant was dialyzed into TEV cleavage buffer (20 mM Tris, 150 mM NaCl, 50 mM imidazole, 2 mM β-mercaptoethanol, pH 7.5) overnight at 4 °C in presence of TEV. WT Vt protein and the Vt variants were then collected and run over Ni-NTA agarose beads. The eluent was collected, concentrated and run over a S100 column (GE, Pittsburg, PA) to obtain the highest level of purity using gel filtration buffer A (10 mM Tris, 200 mM KCl, 10 mM imidazole, 2.5 mM MgCl_2, 1 mM EGTA, 2 mM DTT, pH 7.5) or buffer B (40 mM HEPES, 150 mM NaCl, 2 mM dithiothreitol, pH 7.4). Purified proteins (>96% pure) were concentrated to 100 mM by centrifugation, aliquoted and snap-frozen using liquid nitrogen. Protein stocks were then stored at −80 °C.

## Circular dichroism

To assess the impact of the Vt variants on the conformation and structural stability of Vt, we employed Circular Dichroism (CD) spectroscopy. CD spectra were collected at both near-ultraviolet (350–250 nm) and far-ultraviolet (260–190 nm) spectral regions to monitor secondary and tertiary structural profiles for comparison of WT Vt to the Vt variants using a Jasco J-815 CD spectrophotometer. All spectra were acquired at 20 °C in a buffer containing 10 mM potassium phosphate, 50 mM $Na_2SO_4$, 1 mM dithiothreitol, pH 7.5. Each sample was placed in a 0.1 mm path-length (400 μL) cuvette, and spectra recorded with 0.1 nm data increments at a scanning speed of 50 nm/min. Vt protein concentration of 0.20 mM was used for near-UV and 20 μM for far-UV CD data collection and the resultant spectra averaged over three scans. Thermal melt curves were generated by measuring circular dichroism values at 222 nm over a temperature range of 20 °C to 95 °C, with data collected at intervals of 1 °C/minute. A set of three independent experiments was performed for near UV, far UV and thermal melt curve respectively.

## Lipid co-sedimentation

The following lipids (Avanti Polar Lipids) were employed for lipid cosedimentation assays: (1) 1,2-dioleoyl-*sn*-glycero-3-phosphocholine (PC). (2) 1,2-dioleoyl-*sn*-glycero-3-phosphoethanolamine (PE). (3) 1,2-dioleoyl-*sn*-glycero-3-phospho-L-serine (PS). (4) L-α-phosphatidylinositol-4,5-bisphosphate ($PIP_2$; Brain, porcine). The interaction of WT VT and Vt DAFS variants with lipids was assessed by conducting a co-sedimentation assay with unilamellar vesicles (LUVs)[28,35,85]. Comparison of WT Vt and the Vt DAFS variants to bind PI and PS was assessed using lipid vesicles containing 60% PE, 40% PC by weight, with either $PIP_2$ or PS replacing PE at the concentration indicated. $PIP_2$ binding to Vt variants was characterized using vesicles containing 60% PE, 20% PC, and either 20% PS by weight or $PIP_2$ replacing PC at the concentration indicated. For example, experiments testing the role of 10% $PIP_2$ employed vesicles were composed of 60% PE, 20% PC, 10% PS, and 10% $PIP_2$. The desired quantity of lipid was mixed, dried under nitrogen glass and left in a vacuum overnight. Dried lipids were rehydrated in buffer (40 mM HEPES, 150 mM NaCl, 2 mM dithiothreitol, pH 7.4) for 2 hours with constant shaking, and then extruded with 100 nm polycarbonate membranes in a mini-extruder (Avanti) to form large unilamellar vesicles (LUVs) and stored overnight at 4 °C. The following day, LUVs containing 250 μg of lipids and 10 μl of 100 μM Vt protein (in an identical buffer) were added to each vesicle sample, producing a final volume of 100 μl and incubated for 1 h (at 4 °C, under slow constant rotation). The lipids-protein mixture was spun at 120,000x $g$ in a Beckman TLA100 rotor for 1 h (4 °C). Supernatants and pellets were separated and run on an SDS-PAGE gel, stained with Coomassie Brilliant Blue, analyzed and quantified with ImageJ Software[86].

## Actin co-sedimentation

To assess actin-binding and actin-bundling (crosslinking) properties of WT Vt and the Vt DAFS variants, we performed actin co-sedimentation assays[84]. Briefly, monomeric actin (G-actin) was purified through gel filtration from rabbit muscle acetone powder (obtained from Pel-Freez Biologicals, Rogers, AR) and stored at −80 °C in storage buffer (50 mM imidazole, 100 mM NaCl, 10 mM $MgCl_2$, 10 mM EGTA, 0.5 mM DTT, 0.2 mM ATP, pH 7.0). F-actin was prepared by allowing polymerization of G-actin at 100 μM concentration in actin polymerization buffer (10 mM Tris, 200 mM KCl, 10 mM imidazole, 2.5 mM $MgCl_2$, 1 mM EGTA, 2 mM DTT, pH 7.5) at room temperature under slow constant rotation for 30 minutes. The heterogeneity of F-actin polymers makes it difficult to quantify F-actin concentrations, so the actin concentrations reported here are based on the G-actin concentration. For the actin-binding study, 100 μL samples were prepared by mixing 10 μM Vt variants in actin polymerization buffer and 20 μM F-actin. Samples

were then incubated at room temperature for 1 h and then centrifuged at 185,000x $g$ for 60 min. Similarly, to quantify actin bundling/cross-linking, 100 μL samples were prepared containing 10 μM Vt variants and 20 μM F-actin. The samples were incubated at room temperature under slow constant rotation for 1 h and then centrifuged at 12,000x $g$ for 15 min. For both binding and bundling co-sedimentation assays, the supernatant and pellet were separated by centrifugation, resuspended to equal volumes of SDS-PAGE buffer, and analyzed by 15% SDS-PAGE. F-actin binding and bundling were quantified by determining the fraction of Vt and F-actin, respectively, in the pellets[84]. Densitometry was performed using ImageJ[86].

## Creation of stable cell lines

To generate cell lines that stably express a given Vcn construct, second-generation viral packaging plasmids psPax2 (Addgene Plasmid #12260) and pMD2.G (Addgene Plasmid #12259) were used. To create viral particles, either pRRL-VcnTS (Addgene Plasmid #111830) or a variant thereof, along with psPax2 and pMD2.G plasmids, were co-transfected into HEK293-T cells. Three days post-transfection, media containing viral particles was harvested and stored at −80 °C. One day prior to viral transduction, Vcn$^{-/-}$ MEFs were plated in 6-well dishes at a density of 100,000 cells per dish. Cells were transduced with 500 μL of viral mixture in full media supplemented with 2 μg/mL polybrene (Sigma-Aldrich, St. Louis, USA) to enhance viral uptake. After 3 passages, transduced cells were sorted with flow cytometry into several groups based on intensity of the stable construct's fluorescent signal before being frozen down. An immunofluorescence-based procedure was used to select Vcn$^{-/-}$ MEFs expressing the Vcn biosensor at levels comparable to endogenous Vcn in unaltered MEFs[39].

## Cell culture and transfection

Vcn$^{-/-}$ MEFs and cell lines expressing VcnTS or VcnTS variants were maintained in high-glucose DMEM with sodium pyruvate (Sigma-Aldrich) supplemented with 10% FBS (Cytiva HyClone, Marlborough, MA), 1% v/v non-essential amino acids (Thermo Fisher Scientific, Waltham, MA), and 1% v/v antibiotic-antimycotic solutions (Sigma-Aldrich). HEK293 cells were maintained in high-glucose DMEM (Sigma-Aldrich) supplemented with 10% FBS (Cytiva HyClone) and 1% v/v antibiotic-antimycotic solution (Sigma-Aldrich). Cells were grown at 37 °C in a humidified 5% $CO_2$ atmosphere. For transient transfections, Lipofectamine 2000 (Thermo Fisher Scientific) was added according to manufacturer's instructions.

## Western blot analysis

Cells were washed, lysed with lysis buffer [250 mM NaCl, 10% glycerol, 2 mM EDTA, 0.5% IGEPAL (Sigma-Aldrich), 50 mM HEPES (Sigma-Aldrich)], and then centrifuged at 13,000 RPM for five minutes. Afterwards, 2x Laemmli sample buffer (Bio-Rad Laboratories, Hercules, California) was added to the lysate for a 1:1 dilution and the sample was denatured at 95 °C for five minutes. Then, the sample was loaded onto a 4-20% gradient Mini-PROTEAN TGX precast gel (Bio-Rad Laboratories) and electrophoresed at 100 V for 70 minutes, before being transferred to a PVDF membrane (Bio-Rad Laboratories) via wet transfer. Membranes were blocked with 5% dry milk in TBST [10 mM Tris-HCl, 100 nM NaCl, 0.1% Tween 20] for 1 hour and then incubated with an anti-GFP primary polyclonal antibody (Abcam, ab290) at a 1:5000 dilution or a GAPDH primary polyclonal antibody (Santa Cruz, Dallas, TX, sc25778) at a 1:3000 dilution overnight at 4 °C. Afterwards, the membrane was rinsed three times with TBST and incubated with the appropriate species-specific enzyme-conjugated secondary antibody (Life Technologies), for 1 hour at room temperature. Membranes were later developed using SuperSignal West Pico Chemiluminescent Substrate (Thermo Fisher Scientific). The resulting signal was detected using digital imaging (Bio-Rad Laboratories).

## Cell seeding for fluorescence imaging

For cell imaging, no. 1.5 glass coverslips (Bioptechs, Butler, PA) secured in reusable metal dishes (Bioptechs) were coated overnight at 4 °C with 10 µg/mL FN (Fisher Scientific, Pittsburgh, PA) or 0.01% pLL (Millipore Sigma, Burlington, MA). $Vcn^{-/-}$ MEFs expressing a given Vcn construct were then trypsinized, transferred to the prepared glass-bottom dishes at a density of 100,000 cells per dish and allowed to spread for 1 h on pLL-coated glass and 4 hours on FN-coated glass. For imaging of mature SFs, cells were seeded at 50,000 cells per dish and incubated for 24 h prior to fixation.

To prepare fixed cells, samples were rinsed quickly with PBS (phosphate-buffered saline) and incubated for 10 min with 4% methanol-free paraformaldehyde (Electron Microscopy Sciences, Hatfield, PA). For live cell imaging, growth media was exchanged for imaging media - Medium 199 (Life Technologies, 11043) supplemented with 10% FBS (Cytiva HyClone), 1% v/v nonessential amino acids (Invitrogen), and 1% v/v antibiotic-antimycotic solution (Sigma Aldrich), at least 1 hour before imaging.

## Immunofluorescence staining

Fixed cells were subject to permeabilization in 0.1% Triton-X100 (Millipore Sigma, Burlington, MA) for five minutes at room temperature. Cells were then washed with PBS and then blocked in 2% bovine serum albumin (Sigma-Aldrich) diluted in PBS for 30 minutes. To label F-actin, cells were incubated with Alexa-647 labeled phalloidin (Thermo Fisher Scientific, A22287) at a 1:100 dilution in 2% BSA for 1 hour. Cells were then washed with PBS three times, preceding imaging.

## FRET and fluorescence imaging

An Olympus IX83 inverted epifluorescent microscope (Olympus) equipped with a LambdaLS 300 W ozone-free xenon bulb (Sutter Instruments, Novato, CA), sCMOS ORCA-Flash4.0 V2 camera (Hamamatsu, Japan). motorized filter wheels (Sutter Instruments, 10-3), automated stage (Prior Scientific, H117EIX3), and a FRAPPA (Andor Technologies) equipped with a 514 nm laser was used to image and bleach samples. MetaMorph Advanced software (Olympus, Japan) was used to control the image acquisition and photomanipulation. Unless otherwise indicated, all samples were imaged at 60X magnification (Olympus, UPlanSApo 60X/NA1.35 objective, 108 nm/pix).

Visualization of Vcn in FAs was enabled by quantitation of acceptor intensity signal in the Venus channel with an exposure time of 1000 ms. To visualize F-actin structures, Alexa 647 phalloidin-labeled cells were imaged in the Cy5 channel of the Semrock DA/FI/TR/Cy-5-4X4 M-C Brightline Sedat filter set with an exposure time of 1000 ms.

FRET images were acquired with a three-image sensitized emission acquisition sequence[87,88]. To image mTFP1-Venus sensors, the filter set used includes mTFP1 excitation (Chroma, ET450/30x), mTFP1 emission (Chroma, ET485/20 m), Venus excitation (Chroma, ET514/10x), and Venus emission (Semrock, FF01-571/72) filters and a dichroic mirror (Chroma, T450/514rpc). Images were acquired across the acceptor channel (Venus excitation, Venus emission, 1000 ms exposure), FRET channel (mTFP1 excitation, Venus emission, 1500 ms exposure), and donor channel (mTFP1 excitation, mTFP1 emission, 1500 ms exposure).

To perform FRAP, user-chosen regions of interest were photobleached using a 515-nm laser after taking four prebleach images. To bleach an FA region of interest, 1 laser pulse with a dwell time of 7500 µs per pixel was used. Pre- and postbleach FRAP images were acquired using the aforementioned Venus excitation and emission filters every 5 s until 5 min postbleach, at an exposure time of 250 ms.

## Cell, FA, and cytoplasm segmentation

Individual cells were identified based on closed boundaries drawn by the user on unmasked acceptor channel images. Segmentation of individual FAs was performed using custom-written code in MATLAB

implementing the water algorithm and blob analysis[40,87–89]. The results of FA segmentation were output as a mask and applied across all imaging channels, as well as across images resulting from FRET analysis, including FRET efficiency, for visualization of data. To isolate signals from the cytoplasm, a mask was generated by dilating the FA mask by a scale factor of 10 and then inverting the newly generated mask within cell boundaries. These masks were then applied to acceptor and FRET images of cells with cytoplasmic signal above background pixel intensity.

## Quantification of FA morphometric properties

For each FA, average FRET efficiency, average acceptor intensity and FA area were calculated[90]. To calculate FA morphometric parameters such as FA axis ratio and FA orientation, additional blob analysis methods were applied. Central moments were taken of the pixels ($n$) within a FA $F(x,y)$, about the FA centroid ($\underline{x}, \underline{y}$), to form an equivalent ellipse[91]. Below, $ij$ represents the moment order, where the $ij^{th}$ central moment, $u$, is of order $i+j$. Therefore, the central moment $u_{ij}$ is given by the equation:

$$u_{ij} = \sum_{x=0}^{n} \sum_{y=0}^{n} (x - \underline{x})^i (y - \underline{y})^j F(x,y) \tag{2}$$

Using this equation, the second order central moments ($u_{i+j=2}$) were taken and a covariance matrix, $C_{I(x,y)}$, was subsequently constructed:

$$C_{I(x,y)} = \begin{bmatrix} u_{20} & u_{11} \\ u_{11} & u_{02} \end{bmatrix} \tag{3}$$

FA major ($FA_{maj}$) and minor ($FA_{min}$) axis lengths were then calculated as a function of $C_{I(x,y)}$'s eigenvalues $\lambda_1$, $\lambda_2$ respectively:

$$FA_{maj} = \sqrt{\lambda_1} = \sqrt{2((u_{20} + u_{02}) + \sqrt{(u_{20} - u_{02})^2 + 4(u_{11})^2})} \tag{4}$$

$$FA_{min} = \sqrt{\lambda_2} = \sqrt{2((u_{20} + u_{02}) - \sqrt{(u_{20} - u_{02})^2 + 4(u_{11})^2})} \tag{5}$$

Consequently, FA axis ratio ($FA_{axrat}$) was determined as:

$$FA_{axrat} = \frac{FA_{maj}}{FA_{min}} \tag{6}$$

Lastly, FA orientation from the x-axis was calculated as:

$$\theta_{FA} = \left( \frac{2(u_{11})}{u_{20} - u_{02} + \sqrt{(u_{02} - u_{20})^2 + 4(u_{11})^2}} \right) \tag{7}$$

where $u_{20} > u_{02}$, or

$$\theta_{FA} = \left( \frac{u_{02} - u_{20} + \sqrt{(u_{02} - u_{20})^2 + 4(u_{11})^2}}{2(u_{11})} \right) \tag{8}$$

where $u_{02} > u_{20}$.

## Quantitative FRET efficiency measurements from sensitized emission

For each three-image sensitized emission acquisition, the following process was applied to calculate FRET efficiency. FRET was quantitated on a pixel-by-pixel basis using custom-written code in MATLAB (MathWorks, Natick, MA). Prior to FRET calculations, all images were corrected for dark current and uneven illumination, registered, and

background subtracted[87,88]. FRET imaging of donor-only and acceptor-only samples (cells expressing either a single donor or acceptor FP) were used to calculate spectral bleed-through coefficients[88]. Donor bleed-through coefficients ($d_{bt}$) were calculated for mTFP1 as:

$$d_{bt} = \frac{I_f}{I_d} \qquad (9)$$

where $I_f$ is the FRET-channel intensity and $I_d$ is the donor-channel intensity. Data was binned by donor-channel intensity. Accordingly, acceptor bleed-through coefficients ($a_{bt}$) were calculated for Venus as:

$$a_{bt} = \frac{I_f}{I_a} \qquad (10)$$

where $I_a$ is the acceptor-channel intensity. In this case, data was binned by acceptor-channel intensity. To correct for spectral bleed-through in experimental data, pixel-by-pixel corrected FRET images ($F_c$) were generated according to the equation:

$$F_c = I_f - d_{bt} \times I_d - a_{bt} \times I_a \qquad (11)$$

Upon imaging donor–acceptor fusion constructs of constant high or constant low FRET efficiencies[42], the proportionality constant, $G$, was calculated as:

$$G = - \frac{\Delta\left(\frac{F_c}{I_a}\right)}{\Delta\left(\frac{I_d}{I_a}\right)} \qquad (12)$$

where $\Delta$ indicates the change between two donor-acceptor fusion proteins. Using $G$, FRET efficiency ($E$) was determined:

$$E = \frac{\frac{F_c}{G}}{I_d + \frac{F_c}{G}} \qquad (13)$$

## Quantification of FRAP recovery

Quantification of FRAP recovery was accomplished by tracking acceptor intensity recovery within the bleached FA region of interest[39]. Briefly, user-defined polygons were used to outline an unbleached cytosol control region, an unbleached FA control region and the bleach area region of the FA, on background-subtracted acceptor images. The bleach area region mask was applied to the corresponding cell FRET image to obtain load for the bleached FA region. For each bleach and control region, acceptor intensity was measured in each image taken in the timelapse in ImageJ. FAs that grew, shrank, or moved drastically during the experiment were not analyzed.

The recovery curve was normalized to account for initial intensity and global bleaching. Normalizing for global photobleaching was performed by tracking adhesions that were not photobleached. The normalized recovery curve was then fit to a single exponential recovery equation[39]:

$$Normalized\ Recovery = R_f - (R_f - R_o)e^{-kt} \qquad (14)$$

where $R_f$ is the final recovery and is indicative of the mobile fraction, $R_o$ is the initial recovery, and $k$ is the recovery rate. The half-time of the recovery is determined by $\tau_{1/2} = \ln 2/k$.

## Classification of PR- and SF-associated FAs

To identify focal adhesions associated with protrusions or stress fibers, a tree-based machine learning algorithm was applied to FA-segmented cell images. This machine learning algorithm was trained to differentiate PR- from SF-associated FAs on a ground truth data set of manually identified PR- and SF-associated FAs[41]. Six parameters were identified as necessary to distinguish PR- and SF- associated FAs: FA area, FA axis ratio, 4 nearest neighbors distance, relative orientation, radial orientation, and normalized distance from cell edge[41].

FA axis ratio is defined above as:

$$FA_{axrat} = \frac{FA_{maj}}{FA_{min}} \qquad (15)$$

Relative orientation, describing the orientation of a given FA compared to the average orientation of FAs in a cell of interest, is defined as:

$$\theta_{relative} = \theta_{FA} - \theta_{<FA>} \qquad (16)$$

where $\theta_{<FA>}$ is the average of all FA orientations in a cell of interest and $\theta_{FA}$ is the orientation of a given FA.

FA radial orientation, $\theta_{radial}$, is calculated as:

$$\theta_{radial} = \theta_{subcell} - \theta_{FA} \qquad (17)$$

where $\theta_{subcell}$ is the angle of a vector drawn from the geometric centroid of the cell to the FA of interest.

The FA four nearest neighbors distance, $d_4FA$, which is inversely proportional to FA density, is calculated as:

$$d_4FA = \frac{\sum_{i=1}^{4}\sqrt{(FA_{x,i} - FA_{x,0})^2 + (FA_{y,i} - FA_{y,0})^2}}{4} \qquad (18)$$

where $FA_{x,i}$ and $FA_{y,i}$ represent the FA centroid x and y coordinates, respectively, for each of the four FAs nearest to the FA of interest, $FA_{x,0}$ and $FA_{y,0}$, contained within a cell of interest. The normalized distance from cell edge, $d_{NCE}$, is calculated as:

$$d_{NCE} = \frac{\sqrt{(CE_x - FA_x)^2 + (CE_y - FA_y)^2}}{\sqrt{(CE_x - FA_x)^2 + (CE_y - FA_y)^2} + \sqrt{(CC_x - FA_x)^2 + (CC_y - FA_y)^2}} \qquad (19)$$

where $CE_x$ and $CE_y$ represent the x and y coordinates of the cell edge, $FA_x$ and $FA_y$ represent the x and y coordinates of the FA of interest, and $CC_x$ and $CC_y$ represent the x and y coordinates of the geometric centroid of the cell.

## Analysis of spatial variation in Vcn tension

To detect spatial variations in Vcn tension within a selected population of individual FAs, line scans of single FAs were performed using custom code in MATLAB and JMP Pro software (Statistical Analytical Systems). For a given FA, the improfile.m MATLAB function was used to collect FRET and acceptor intensity pixel values, along the FA major axis length, with a line thickness of minor axis length/2, for each FA within a cell. A cubic spline-based smoothing algorithm was used for noise reduction of the acceptor intensity and FRET efficiency profiles[41].

To probe all the identified FAs in each class, spatial variations in Vcn tension were detected by comparing the geometric centroid, which provides information on the physical location of the FA, to the FRET$^{-1}$-weighted centroid, which is shifted towards areas of higher tension.

The geometric cell centroid is calculated as x- and y coordinates,

$$x_{geo} = \frac{1}{n}\sum_{i=1}^{n} x_i \qquad (20)$$

$$y_{geo} = \frac{1}{n}\sum_{i=1}^{n} y_i \tag{21}$$

where $x_{geo}$ and $y_{geo}$ represent the x- and y-coordinates corresponding to the $i^{th}$ pixel location for all $n$ pixels within a given FA. The $FRET^{-1}$-weighted centroid is calculated as x- and y-coordinates, $x_{FRET^{-1}}$ and $y_{FRET^{-1}}$

$$x_{FRET^{-1}} = \frac{\sum_{i=1}^{n}(I_{i,FRET})^{-1}x_i}{\sum_{i=1}^{n}(I_{i,FRET})^{-1}} \tag{22}$$

$$y_{FRET^{-1}} = \frac{\sum_{i=1}^{n}(I_{i,FRET})^{-1}y_i}{\sum_{i=1}^{n}(I_{i,FRET})^{-1}} \tag{23}$$

where $I_{i,FRET}$ is the FRET index at the $i^{th}$ pixel location for all $n$ pixels within a given FA. To maintain a relationship with tension, the inverse of $I_{i,FRET}$ was used. As FAs are randomly oriented and positioned within an imaging frame, we also calculate a cell centroid to use as a frame of reference. The cell centroid is calculated as:

$$x_{cell} = \frac{1}{N}\sum_{i=1}^{N} x_{geo} \tag{24}$$

$$y_{cell} = \frac{1}{N}\sum_{i=1}^{N} y_{geo} \tag{25}$$

for a cell of N FAs. A spatial variation index (SVI) was calculated as:

$$SVI = \frac{\sqrt{(x_{FRET^{-1}}-x_{cell})^2 + (y_{FRET^{-1}}-y_{cell})^2} - \sqrt{(x_{geo}-x_{cell})^2 + (y_{geo}-y_{cell})^2}}{FA_{maj}} \tag{26}$$

where $FA_{maj}$ represents the FA major axis length. This normalization was used to account for differences in FA size. This index is defined such that more distally skewed (away from the cell center) Vcn tension profiles are represented as larger positive values.

## Quantification of SF orientation

Custom written code in MATLAB was used to identify and analyze actin SFs. Briefly, the structure tensor $S_0$ of each SF pixel was computed to determine maximal gradients of actin intensity and the orientation of SFs on a pixel-by-pixel basis[92].

Image I($p$), where each pixel $p$, at position ($x,y$) has the structure tensor:

$$S_0(p) = \begin{bmatrix} (I_x(p))^2 & I_x(p)I_y(p) \\ I_x(p)I_y(p) & (I_y(p))^2 \end{bmatrix} \tag{27}$$

The eigenvectors, $\mathbf{e_1}$ and $\mathbf{e_2}$, and eigenvalues, $\mathbf{l_1}$ and $\mathbf{l_2}$, of each pixel's structure tensor were determined. For each pixel, the angle formed by the components of the larger eigenvector ($\mathbf{e_1}$) was calculated in radians as the SF orientation ($\theta_{SF}$):

$$\theta_{SF} = \left(\frac{e_{1y}}{e_{1x}}\right) \tag{28}$$

Images of $\theta_{SF}$ and $\mathbf{l_1}$ (magnitude of the actin intensity gradient) were consequently generated. SF masks were generated by isolating pixels of the $\mathbf{l_1}$ image above a desired actin intensity magnitude[93]. SF masks were applied to the $\theta_{SF}$ image to isolate orientations of individual SFs. The standard deviation of SF orientations per cell was calculated to assess the uniformity, or lack thereof, of SF orientations in the cell.

## Quantification of coalignment of FAs and SFs

To probe the coalignment of FAs and SFs, we first identified the local regions most likely to be associated with a given FA by creating a Voronoi tessellation[94] based on the centroid position of the FAs in VcnTS or VcnTS E1015A-E1021A expressing Vcn$^{-/-}$ MEFs that had also been labeled with phalloidin to visualize F-actin structures. Within each Voronoi region, local SF density (measured as the percent area of SFs in a Voronoi region) was quantified. For each cell, the local SF densities of all the Voronoi regions were averaged. To obtain a representative SF orientation per local region, each cell's Voronoi tessellation mask was applied to $\theta_{SF}$ images and a single vector corresponding to SF orientation per region was calculated. From blob analysis, the orientation of each Voronoi region's respective focal adhesion was obtained. Relative orientation ($\theta_{Rel}$) of a local averaged SF orientation to its respective FA orientation was calculated:

$$\theta_{Rel} = |\theta_{SF} - \theta_{FA}| \tag{29}$$

$\theta_{Rel}$ for each local region was then averaged per cell.

## Directed cell migration assay

Transwell inserts with 8-μm pores (Corning, Corning, NY) were prepared with fibronectin[39]. Briefly, 10 μg/mL of fibronectin solution was used to coat the underside of the transwell insert, and the insert was then placed in serum-free media. Cells were serum-starved for 2 hours prior to seeding. For the seeding procedure, -16,000 cells in serum-free media were plated into the upper chambers of each well. Cells were allowed to migrate for 4 hours at 37°C. After migration, cells were fixed in 4% methanol-free paraformaldehyde and permeabilized in 0.1% Triton X-100 solution. Using a cotton swab, non-migratory cells from the upper chamber were gently removed. The remaining cells on the underside were stained with Hoechst (Thermo Fisher Scientific) to visualize nuclei. The chambers were subsequently washed with 1x PBS. The chambers were then imaged at 10x on the previously mentioned epi-fluorescence microscope. To analyze the images, rolling ball background subtraction was performed in ImageJ (NIH), and cell nuclei were counted per image frame via the Analyze Particles tool.

## Statistical analysis

All experimental data sets were initially assessed using Levene's for unequal variances. For two sample data sets, a one-way ANOVA followed by an F-test was performed when comparing two means with equal variances, and a Welch's t-test was used when comparing two means with unequal variances. For multi-sample data sets, a one-way ANOVA followed by post-hoc Tukey HSD (equal variances) or post-hoc Steel-Dwass test (unequal variances) was performed for multiple comparisons using Student Version of Origin software (OriginLab Corporation, MA-01060, USA) or JMP Pro (SAS Institute, NC-27519, USA). Bar graph data is presented as mean ± S.E.M. unless otherwise noted. Box depicts the median as the center value, the 25$^{th}$ percentile as the lower bound, and the 75$^{th}$ percentile as the upper bound. Whiskers extend 1.5 times the IQR from the bottom and top of box, or to the minimum and maximum of the data if the data does not extend to the whiskers. Values outside the whiskers are plotted as individual points. The values for N, n, p, and statistical test performed are listed in the figure captions. Details regarding two-sample statistical comparisons are presented in captions and multi-sample statistical comparisons are presented in Supplementary Tables 1–20.

## Reporting summary

Further information on research design is available in the Nature Portfolio Reporting Summary linked to this article.

## Data availability

PDB ID: 6UPV (Alpha-E-catenin ABD-F-actin complex))[65]. PDB ID: 3JBI (vinculin tail domain bound to F-actin)[29]. PDB ID: 1QKR (vinculin tail))[78]. The co-ordinates of force-loaded and unloaded states of Vt:F-actin complex in the $F_{pointed}$ direction are provided as supplementary data files. Figshare https://doi.org/10.6084/m9.figshare.23686710. All remaining data are available in the Article, Supplementary, and Source Data files. All data reported in this paper will be shared by the lead contacts upon reasonable request. Source data are provided with this paper.

## Code availability

All image analysis code has been deposited at https://gitlab.oit.duke.edu/HoffmanLab-Public and is publicly available as of the date of publication. Any additional information required to reanalyze the data reported in this paper is available from the lead contacts upon reasonable request.

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

## Acknowledgements

The authors thank James E. Bear for providing critical input and discussion. Illustrative images in Figs. 1, 4, Supplementary Figs. 3, 4, 5, and 19 were created with BioRender.com. Illustrative images in Figs. 5, 6, 8, Supplementary Figs. 1, 7, 8, 11, and 16 were created with Adobe Illustrator. Research reported in this publication was supported by National Institutes of Health (NIH) grants R01GM115597 and R35GM134962 (to S. L. Campbell). National Institutes of Health (NIH) grants 1R35GM134864, and 1RF1AG071675, National Science Foundation (NSF) grant 2210963, and the Passan Foundation (to N. V. Dokholyan). National Institutes of Health (NIH) grant R01GM121739 and National Science Foundation CAREER Award NSF-CMMI-14-54257 (to B. D. Hoffman). National Science Foundation (NSF) Graduate Research Fellowship Program (GFRP) 2139754 Award (to J. N. Malavade).

## Author contributions

Conceptualization: V.R.C., M.A.I.K., J.N.M., N.V.D., B.D.H., S.L.C. Methodology: V.R.C., M.A.I.K., J.N.M. Simulation: V.R.C. Experimentation: M.A.I.K, J.N.M. Analysis: J.N.M., V.R.C., M.A.I.K. Visualization: V.R.C., M.A.I.K., J.N.M. Supervision: B.D.H., N.V.D., S.L.C. Writing—original draft: V.R.C., M.A.I.K., J.N.M., B.D.H., S.L.C. Writing—review & editing: B.D.H., S.L.C., N.V.D. N.V.D., B.D.H., and S.L.C. contributed equally to this work.

## Competing interests

The authors declare no competing interests.
