## [Peer review file · Nature Communications]

REVIEWER COMMENTS

Reviewer #1 (Remarks to the Author):

In the manuscript “Elucidation of the molecular basis and cellular functions of vinculin-actin directional catch bonding”, a variety of cellular biology, molecular, biophysical and in silico simulation methods were applied with the aim to dissect the molecular mechanisms behind the mechanical anisotropy of the Vinculin-Actin complex catch bond. The work identifies directionally asymmetric, force strengthening (DAFS) residues and demonstrate their contribution to the ability of Vinculin to bear load and mediate catch bonding.

The authors also present a suite of tools for studying catch bonding in biological systems. Within this framework, the DAFS variants are extensively verified for interference in several relevant aspects, including Vinculin (Vcn) activation, protein structure, actin binding, actin crosslinking, lipid binding, morphological aspects, and haptotaxis.

The data is clear and convincing and may be particularly relevant for studying directed cell migration. The manuscript is mostly well-written and compelling. Regrettably, the authors apparently mishandled some figure numbers. As specifically mentioned, below, Figures 8 and 9 were cited but not included in the manuscript.

I believe that the manuscript could be published at Nature Communications after a few problems were addressed.

Major points:

Could the authors comment if the analysis of the umbrella sampling simulations suffices to explain the catch bond directionality based on the H-bonds results?

To enhance the understanding of physiologically relevant modes of Vcn:F-actin engagement and better assess their binding under different loading directions, it would be worthwhile for the authors to add some analysis. For the H-bonds, it would be interesting to see what is the fraction of native H-bonds that is preserved. Since pulling windows and H-bond are discrete, one could represent the H-bonds along the windows by pair. With less details the average number of H-bonds/window over a 2D density map to encompass force and distance.

Also, it is undeniable that H-bonds play an important role in this complex, but one shall not ignore the remaining contacts. Could the authors comment about how does the contact area between Vt and F-actin behaves in three pulling directions in comparison to the native state? In addition to the differential H-bonds is there an increase in the area due to novel contacts or tightening of native interactions? Can the authors also identify additional relevant residues?

Finally, to support the generalizability argument in the discussion it would be beneficial to provide a supplementary figure showing the structural alignment of Catn and Vt indicating the residues Catn:D813,M816 and Vt:E1015,E1021.

About the figures

Numbering of supplementary tables is confusing, tables are labeled "Figure" or "Supp Figure".

At figure 5, and figure S2, I found the "A" and "B" letters on top of boxplot quite confusing, are they supposed to mean $p < 0.05$? Authors may use a different symbol.

Figure S3 is never mentioned in the main text. It appears that at line 299 authors meant to reference Figure S3 instead of S2.

Figure 8, cited at line 314 was not included or has a different number.

Figure 9, cited at line 317 was not included or has a different number.

Figure S8, cited at line 337 deals with FRET efficiency instead of the local orientation of SFs.

Supporting figure title "Figures S10. Figure 8" is confusing. It is not cited in the main text.

Methods

Clarify the system setup for 3JBI, it appears at the two first topics. At the second one it contains additional model refinement steps.

During pulling simulations position restraints were applied to the backbone atoms of F-actin but there is no mention to which atoms from Vt were selected to pull.

The Vt:F-actin structure (PDB ID:3JBI) is mentioned 6 times. It is sufficient to mention it once in the main text and where relevant at figure caption.

minor comments

Line 55: (FAs) appears in parenthesis without prior abbreviation.

Line 56: cell-ECM (Extracellular Matrix)

Lines 86/87: Please add citation to “Knockout of Vcn in mice causes cardiac and neural tube developmental defects and embryonic lethality at ~ E10”

Line 168: Parameters for H-bonds distance and angle are already defined at the Methods section.

Line 493/494: “position restraints energy minimization for 100 ps under constant-temperature, constant-volume (NVT) ensemble.”

Minimization does not involve time; it might be replaced by MD simulation or equilibration.

Line 525: Replace “h-bonds” by H-bonds

Line 527: Replace “Gromacs” by GROMACS

Line 528: Replace “PyMol” by PyMOL

Reviewer #3 (Remarks to the Author):

In their manuscript, Venkat and colleagues developed a computational approach which predicted catchbonding variants of vinculin. These variants, when introduced in tension sensors reduced vinculin loading which was consistent with models suggesting loss of directional catch bonding. Mutants also perturbed F-actin alignment and directed cell migration. I have no expertise in modelling structural dynamics and cannot comment on that part. In general, the combination of modelling with cell tension sensors is exciting and of interest to the Nat Comms readership. However, the sensor experiments are not entirely convincing and the functional cell assays seem little appropriate to investigate vnc catchbonds. I feel this part needs major revision before publication.

Main points:

1) I am not certain why F-actin arrangements and migration were measured for the different sensors. Introducing forces that induce actin rearrangements (e.g. shear flow or cell stretching) to me seems a much better way of studying the probes. Changes in cell migration or actin rearrangements over longer periods could be due to other signalling influences (particularly in cells where probes are overexpressed). Changes in FRET before and after the application of force would be a powerful way to

show the differences between the various TS probes. Since they are intramolecular FRET probes it should be fairly easy to record changes in the same cell under different force regimes.

2) Ideally, authors should use FLIM to determine FRET efficiencies, which is much more reliable (PMID: 23203070 and PMID: 30865383) than sensitized emission. The accuracy in sensitized emission has strong limitations because of the bleedthrough parameters and corrections for chromatic (and magnification) aberrations that need to be corrected for.

3) Whilst I like the thorough quantification of the sensor experiments, it would be better to show the complete image for each FRET construct, i.e. FAs together with the cytoplasm to see the relation between the two. Separate images can show the masks used for measurements and/or show the images for the separate areas.

4) All the images for quantification of the cytoplasmic pool of VncCS (Fig.S5) and VncTS sensors (Fig.S8) show a steep gradient of FRET decrease towards the cell edge. This does raise some concerns and one wonders whether the sensors do what they claim to do. What is the reason for this gradient? Were these measurements done in fixed or live cells?

5) There is no negative or positive control for the VcnCS sensor, and therefore it is unclear whether it works. Mutations published for a constitutively active Vcn (T12) would help.

6) The TSMOD seems rather low in FRET (at least the image) in comparison with the cytoplasmic fraction of the various TS sensors (Fig.S8). Were they differently scaled or are the differences due to differences in the rotational freedom or other properties of the probes?

REVIEWER 1 COMMENTS:

Issue 1.1, Provide more detail on the interpretation of the umbrella sampling simulation:

→ **Reviewer Comment:** *“Could the authors comment if the analysis of the umbrella sampling simulations suffices to explain the catch bond directionality based on the H-bonds results?”*

Response: We appreciate the reviewer's question regarding the sufficiency of umbrella sampling simulations in explaining the directionality of catch bonds based on H-bond results. Catch bond behavior is a complex phenomenon influenced by various factors beyond H-bonds alone. Our initial analysis of the umbrella sampling simulations has provided valuable insights into the energetics and stability of the Vcn:F-actin complex under different pulling directions, specifically regarding the formation and stability of H-bonds between Vcn and F-actin residues. However, we acknowledge that catch bond directionality is governed by a combination of interactions, including H-bonds, hydrophobic interactions, salt bridges, and van der Waals interactions.

To gain a more comprehensive understanding of catch bond directionality, we performed additional simulations using constant-force pulling discrete molecular dynamics (DMD) simulations on the F-actin:Vt complex. These DMD simulations allowed us to explore the properties and atomistic details of directionally asymmetric force-strengthening (DAFS) interactions between Vt and F-actin. By subjecting the complex to pulling forces in different directions relative to F-actin, the DMD simulations captured force-induced conformational changes, allosteric effects, and additional non-covalent interactions that contribute to catch bond behavior.

The combination of umbrella sampling and DMD simulations enhances our understanding of catch bond directionality by considering both the energetic landscape and the dynamic behavior of the complex under force. Together, these simulations provide valuable insights into the intricate interplay of structural changes and molecular interactions that govern the directionality of catch bonds between Vt and F-actin.

→ **Action:** We have included a paragraph on page no. 11 for clarity.

Issue 1.2, Further analysis of H-bond formed as function of force and direction:

→ **Reviewer Comment:** *“To enhance the understanding of physiologically relevant modes of Vcn:F-actin engagement and better assess their binding under different loading directions, it would be worthwhile for the authors to add some analysis. For the H-bonds, it would be interesting to see what is the fraction of native H-bonds that is preserved. Since pulling windows and H-bond are discrete, one could represent the H-bonds along the windows by pair. With less details the average number of H-bonds/window over a 2D density map to encompass force and distance.”*

→ **Response:** We thank the reviewer for suggesting additional analyses to enhance the understanding of the physiologically relevant modes of Vcn:F-actin engagement and to better assess binding interactions under different loading directions. To address these concerns, we first analyzed the center of mass (CoM) distance between Vt and F-actin as function of force amplitude and force direction by constructing a variety of 2D density maps (Fig. S3). Excitingly, these analyses reveal the increased formation of H-bonds only when forces are applied in the F_{pointed} direction. Furthermore, we assessed the fraction of native Vt:F-actin H-bonds that are retained under different loading conditions (Fig. S3D). This analysis demonstrates that directional catch bonding is partially due to stabilization of native H-bonds. To illustrate the relative H-bond occupancies of native and non-native H-bonds as a function of applied forces, we generated circular plots of H-bond occupancy over the regime where dissociation of the Vt:F-actin complex occurs, 124-132 pN, in all three loading directions (Fig. 1C, S4). This analysis revealed that the non-native H-bonds mediated by Vt residues E1015 and E1021, have particularly high occupancy and exhibit substantial force-strengthening in the F_{pointed} direction. We then assessed changes in H-bond formation for the Vt catch bond deficient variant, E1015A-E1021A, in response to forces applied in the F_{pointed} direction (Fig S5D). This analysis revealed that formation of H-bonds by E1015 and E1021 in Vt is critical for maintaining a number of intermolecular interactions (including contacts other than H-bonds, see Issue 1.3 for further information) that contribute to the overall strength of Vt:F-actin directional catch bonding. A figure depicting how force application in the F_{pointed} direction affects the Vt:F-actin complex that highlights the key role of Vt residues E1015 and E1021 in directional catch bonding has also been added to the manuscript (Fig S5D).

→ Action:

- A figure summarizing an analysis of the relationship between H-bond formation, distance between Vt and F-actin, and applied force in various directions has been added to the manuscript (Fig.S3).
- A figure panel showing the preservation of native H-bonds as a function of force amplitude and force direction has been added to the manuscript (Fig.S3D)
- Figures summarizing H-bond formation as a function of force amplitude and force direction with more intuitive circular plots have been added to the manuscript (Fig.1C, Fig.S4).
- A figure panel assessing H-bond formation as a function of force amplitude and force direction for the catch bond deficient E1015-E1021 variant has been added to the manuscript (Fig.S5D).
- A figure depicting how forces applied in the $F_{pointed}$ direction, affect the role of native and non-native H-bonds has been added (Fig. S18).
- The cartoon summarizing our working model of forces applied in different directions has been modified to precisely depict the role of native and non-native H-bonds (Fig. S19).
- The manuscript has been extensively edited to describe the importance of Vt residues E1015 and E1021 in mediating directional catch bonding through formation of non-native H-bonds and stabilization of intermolecular interactions within the Vt:F-actin complex.

Fig. 1 (C) A circular plot depicting Vt and F-actin interactions in the $F_{pointed}$ direction to probe Vcn:F-actin engagement. The occupancy percentages of inter-molecular H-bonds between F-actin and Vt as quantified from constant-force pulling DMD simulations were used to create the circular plot. A scatter plot was used to illustrate the occupancies of H-bonds, allowing for a detailed analysis of the H-bond occupancies at specific forces. Each dot on the plot represents the existence of H-bonds at different pulling forces, color-coded according to force magnitude. The size of each dot indicates the occupancy percentage of the corresponding H-bond during the pulling simulation at a specific force. Native H-bonds observed in the cryo-EM structure of the Vt:F-actin complex (PDB: 3JBI) are highlighted by black triangles. DAFS H-bonds that signify directional catch bonding between Vt and F-actin are illustrated as blue ribbons while normal H-bonds are represented by grey ribbons.

Fig. S3. H-bond density plots and fraction of native H-bonds from DMD pulling simulations in three pulling directions.

Analysis of H-bonds between Vt and F-actin as a function of different pulling directions. Density maps of the average number of H-bonds as a function of force and Vt:F-actin center of mass (CoM) distance are shown in Panels A-C, highlighting strong and weak interactions between the two molecules. The engagement of DAFS interactions results in a higher number of H-bonds between Vt and F-actin in the F_{pointed} direction. Panel D shows the fraction of native H-bonds preserved between F-actin and Vt in the various pulling directions and loading conditions, as calculated by the ratio of the number of H-bonds present in the pulled state to the number of H-bonds in the unloaded state (PDB: 3JBI). H-bonds with occupancy > 10% were employed for evaluation of the fraction of native H-bonds.

Fig. S4. Occupancy percentages of inter-molecular H-bonds between Vt and F-actin.

To investigate Vcn:F-actin engagement in the **(A)** F_{barbed} and **(B)** F_{normal} directions, circular plots of Vt (red) and F-actin (black) interactions were generated. The occupancy percentages of intermolecular H-bond interactions between F-actin and Vt obtained from constant-force pulling DMD simulations were used to create the circular plots. Each dot on the scatter plot represents the presence of an H-bond interaction at different pulling forces, with the color indicating the magnitude of the applied force. The size of each dot corresponds to the occupancy percentage of the respective H-bond during the pulling simulation at a specific force. Native H-bonds observed in the unloaded state based on the cryo-EM structure of the Vt:F-actin complex (PDB: 3JBI) are depicted as black triangles on the heatmap with the intermolecular H-bonds visualized as grey ribbons, providing a representation of the spatial arrangement of these interactions.

Fig. S5. DMD simulations predict that the Vt E1015A-E1021A variant abolishes most DAFS interactions.

To validate the presence of DAFS interactions, a Vt variant (E1015A-E1021A) was introduced into the Vt:F-actin complex, and its engagement with F-actin in the F_{pointed} direction was investigated through constant-force pulling DMD simulations. Quantification of the **(A)** contact area, **(B)** average number of salt-bridge interactions, and **(C)** average number of vdW interactions between Vt_E1015A-E1021A and F-actin in the F_{pointed} direction. A cut-off distance of 3.5 Å and 5 Å was employed to evaluate salt-bridge and vdW interactions, respectively, using VMD. The *gmx sasa* tool was used to plot contact area versus pulling force. Pulling force in the range of 126 – 128 pN is highlighted to indicate the DAFS interaction force regime. **(D)** Circular plot illustrating intermolecular H-bond occupancies between the Vt_E1015A-E1021A variant (red) and F-actin (black) at different pulling forces is shown. Each dot on the plot represents a H-bond interaction, with the color of the dot indicating the magnitude of the applied force and the size of the dot the H-bond occupancy percentage. Native H-bond interactions observed in the unloaded state (based on the cryo-EM structure of the Vt:F-actin complex (PDB: 3JBI)), are depicted as black triangles on the heatmap. The H-bond interactions are visualized as grey ribbons, providing a representation of the spatial arrangement of these interactions.

Fig. S18. F_{pointed} -directed pulling force induces structural changes in the Vt helical bundle to strengthen existing and new catch bonds between F-actin and Vt.

DMD pulling simulation trajectories depict twisting and conformational reorientation of helix-4 and helix-5 of Vt in the F_{pointed} direction that exposes side chains of E1015 and E1021 to generate a new actin binding interface. These structural changes also reinforce existing H-bonds between helix-4 of Vt and F-actin. Color scheme: Unloaded Vt in light gray, force-loaded Vt in red, and F-actin in dark gray. The catch bond forming residue pairs are shown in licorice representation.

Fig. S19. A schematic representation of directional catch bonding in Vcn.

Force-induced structural changes in Vt facilitate formation of directionally asymmetric catch bonds with F-actin. Specifically, F_{pointed} directed pulling force in the range 124 – 128 pN induces a twisting rearrangement of the Vt helical bundle along the short-axis of the actin filament. This twisting motion generates a new binding interface at the C-terminal end of helix-4 and helix-5 that promotes formation of new H-bond interactions as well as strengthens existing H-bonds. In contrast, the strength of existing H-bonds in F_{barbed} trajectories was reduced relative to F_{pointed} , whereas only weak and transient interactions were observed in F_{normal} trajectories.

Issue 1.3, Discussion of types of contacts other than H-bonds:

➔ **Reviewer Comment:** “Also, it is undeniable that H-bonds play an important role in this complex, but one shall not ignore the remaining contacts. Could the authors comment about how does the contact area between Vt and F-actin behaves in three pulling directions in comparison to the native state? In addition to the differential H-bonds is there an increase in the area due to novel contacts or tightening of native interactions? Can the authors also identify additional relevant residues?”

→ **Response:** Thank you for your insightful comment. We agree that while H-bonds play an important role in the Vt:F-actin complex, other types of intermolecular contacts should not be overlooked. To address this question, we analyzed the contact area, salt bridges, and Van der Waals interactions (cut-off 5Å) between Vt and F-actin in three pulling directions (Fig.S2, shown below for convenience). This analysis revealed that these types of intermolecular interactions are preferentially maintained, but not strengthened, by force application in the F_{pointed} direction. Furthermore, maintenance of these intermolecular interactions was ablated in the E1015A-E1021A Vt variant (Fig S5A-S5C). In combination with the data discussed in Issue 1.2, these results demonstrate a key role for Vt residues E1015 and E1021 in directional asymmetric force sensitive Vt:F-actin interactions.

→ **Action:**

- A figure describing the average contact surface area between the Vt and F-actin, the number of salt-bridges, and the number of Van Der Waals interactions change as a function of applied force application and direction, has been added (Fig. S2).
- A figure panel has been added highlighting loss of DAFS H-bonds for the Vt E1015A-E1021A variant with forces are applied in the F_{pointed} direction (Fig. S5).
- A figure has been added depicting how forces applied in the F_{pointed} direction induce structural changes in the Vt helical bundle to strengthen existing and new catch bonds between F-actin and Vt, (Fig. S18).
- The cartoons summarizing our working model of forces applied in different directions has been edited to more precisely depict the role of maintenance of Vt:F-actin interface and various types of intermolecular interactions in directional catch bonding (Fig. S19).
- The manuscript has been extensively edited to discuss the importance of Vcn residues E1015-E1021 in mediating directional catch bonding through the stabilization of intermolecular interactions within the Vt:F-actin complex.

Fig. S2. Evaluation of contact area, salt-bridges, and van der Waals (vdW) interactions between Vt and F-actin.

Quantification of the (A) contact area, (B) average number of salt-bridge interactions, and (C) average number of vdW interactions between Vt and F-actin from constant-force pulling DMD simulations in F_{pointed} , F_{barbed} , and F_{normal} directions. A cut-off distance of 3.5 Å and 5 Å was employed to evaluate salt-bridge and vdW interactions, respectively, using VMD. The sidechains of charged residues exhibit dynamic behavior and undergo transient interactions, resulting in higher uncertainty and variability. The *gmx sasa* tool was used to plot contact area versus pulling force. Pulling force range in the range of 126 – 128 pN is highlighted to indicate the DAFS interaction force regime.

Issue 1.4, Strengthening argument for generalizability to alpha-catenin:

→ **Reviewer Comment:** “Finally, to support the generalizability argument in the discussion it would be beneficial to provide a supplementary figure showing the structural alignment of Catn and Vt indicating the residues *Catn:D813,M816* and *Vt:E1015,E1021*.”

→ **Response:** We agree this will strengthen the paper and we thank the reviewer for this suggestion.

→ **Action:**

- The requested figure has been added as Fig. S20.
- Additionally, we updated the discussion associated with this figure (page no: 12-13) to highlight similarities/differences between the Vcn:F-actin and Catn:F-actin catch bond interactions.

Fig. S20. Structural alignment of Catn:F-actin and Vt:F-actin complexes.

Structural superimposition of Catn:F-actin (PDB ID: 6UPV) and Vt:F-actin complex (PDB ID: 3JBI) cryo-EM reconstructions highlighting key residues involved in intermolecular interactions. Catn is shown in green, Vt in red, F-actin in gray, and the ADP nucleotide in blue. Key residues involved in intermolecular interactions, Catn: D813, M816 and Vt: E1015, E1021, are shown as stick models.

Issue 1.5, Clarity of statistics presentation:

→ **Reviewer Comment:** “At figure 5, and figure S2, I found the “A” and “B” letters on top of boxplot quite confusing, are they supposed to mean $p < 0.05$? Authors may use a different symbol.”

→ **Response:** We apologize for the confusion. This is a standard nomenclature used in some fields where a large number of comparisons are often made. As we compared many constructs to each other, we have adopted this formalism. Datasets marked with the same letter are not statistically significant. Datasets marked with different letters are statistically different at $p < 0.05$, with corrections for multiple comparisons as appropriate.

→ **Action:**

- We have updated the text to better describe data statistics in the figure captions (See Fig. 5, Fig. 6, and Fig. 8).
- The figure labels are now placed outside of the plots and do not interfere with indications of statistical significance.

Issue 1.6, Errors with figure naming and numbering:

→ **Reviewer Comment:** There were a large number of errors in figure naming and numbering. (Paraphrased for brevity)

→ **Response:** We apologize for the excessive number of errors and thank the reviewer for the detailed reading of the manuscript.

→ **Action:**

- We have extensively edited the manuscript for clarity and accuracy. We have corrected all errors in figure numbering, including changes associated with the resubmission.

Issue 1.7, Clarity of the presentation of methods used in simulations:

→ **Reviewer Comment:** “Clarify the system setup for 3JBI, it appears at the two first topics. At the second one it contains additional model refinement steps. During pulling simulations position restraints were applied to the backbone atoms of F-actin but there is no mention to which atoms from Vt were selected to pull.”

→ **Response:** We appreciate the reviewer's comment regarding the clarification of the system setup for 3JBI and the details regarding the pulling simulations. We have now provided a thorough description of the system setup for 3JBI including the preparation of the F-actin and Vt components. We have added the following information regarding additional model refinement steps.

During the pulling simulations, we applied position restraints specifically to the backbone atoms of F-actin and chose the center of mass (CoM) of Vt as a reference point for applying external forces .in three different directions (F_{pointed} , F_{barbed} , and F_{normal}) to investigate the effects of force on the Vt:F-actin complex. Specific atoms within Vt were not individually restrained, but rather the CoM of the entire Vt molecule was used as the point of force application.

→ **Action:**

- Text describing the Vt pulling position during DMD simulations has been added to the methods section.

Issue 1.8, Fixing of typographical errors, missing citations, and undefined abbreviations:

→ **Reviewer Comment:** *There were many such errors in the manuscript (paraphrased).*

→ **Response:** We thank the reviewer for the detailed reading of our manuscript.

→ **Action:**

- We have extensively edited the manuscript for clarity and accuracy. We have done our best to remove the errors noted by the reviewer, as well as any others we found in the process of revising the manuscript.

→ **Reviewer Comment:** *The Vt:F-actin structure (PDB ID:3JBI) is mentioned 6 times. It is sufficient to mention it once in the main text and where relevant at figure caption.*

→ **Action:** Deleted multiple entries of PDB code.

→ **Reviewer Comment:** *Line 55: (FAs) appears in parenthesis without prior abbreviation.*

→ **Action:** Expanded FA in line 55.

→ **Reviewer Comment:** *Line 56: cell-ECM (Extracellular Matrix)*

→ **Action:** Expanded ECM in line 56.

→ **Reviewer Comment:** *Lines 86/87: Please add citation to "Knockout of Vcn in mice causes cardiac and neural tube developmental defects and embryonic lethality at ~ E10"*

→ **Action:** The following citation has now been added. Xu W, Baribault H, Adamson ED. Vinculin knockout results in heart and brain defects during embryonic development. *Development*. 1998 Jan;125(2):327-37. doi: 10.1242/dev.125.2.327. PMID: 9486805.

→ **Reviewer Comment:** *Line 168: Parameters for H-bonds distance and angle are already defined at the Methods section.*

→ **Action:** Deleted this repetitive statement.

→ **Reviewer Comment:** *Line 493/494: "position restraints energy minimization for 100 ps under constant-temperature, constant-volume (NVT) ensemble." Minimization does not involve time; it might be replaced by MD simulation or equilibration.*

→ **Action:** The sentence has now been modified as follows:

We chose the CHARMM36 forcefield to generate bonded and non-bonded interaction parameters for all atoms in the system, and performed steepest descents energy minimization followed by position restraints equilibration for 100 ps under constant-temperature, constant-volume (NVT) ensemble.

→ **Reviewer Comment:** *Line 525: Replace "h-bonds" by H-bonds*

→ **Action:** This has now been corrected.

→ **Reviewer Comment:** Line 527: Replace “Gromacs” by GROMACS

Action: This has now been corrected.

→ **Reviewer Comment:** Line 528: Replace “PyMol” by PyMOL

→ **Action:** This has now been corrected.

REVIEWER 2 COMMENTS:

Issue 2.1, Probing response to applied forces

→ **Reviewer Comment:** *“I am not certain why F-actin arrangements and migration were measured for the different sensors. Introducing forces that induce actin rearrangements (e.g., shear flow or cell stretching) to me seems a much better way of studying the probes. Changes in cell migration or actin rearrangements over longer periods could be due to other signaling influences (particularly in cells where probes are overexpressed). Changes in FRET before and after the application of force would be a powerful way to show the differences between the various TS probes. Since they are intramolecular FRET probes it should be fairly easy to record changes in the same cell under different force regimes.”*

→ **Response:** This comment has three parts, which we will address in the following order: lack of clarity on why studies of F-actin rearrangement and cell migration, potential for over-expression effects, and a lack of experiments assessing the effects of external force application.

We regret that the intention of the experiments focusing on F-actin arrangements and migration was not clearly explained in the main text. The purpose of these experiments is not to “test the sensors”, but to show the effects of the DAFS variants on subcellular structures and cell behaviors. We note that both vinculin tension sensors (VcnTSs) and vinculin conformation sensors (VcnCSs) have been used for over a decade by us and others and the sensors are functionally indistinguishable from WT Vcn¹⁻⁶. We also note that we have previously used VcnTSs harboring point mutations expressed in Vcn^{-/-} MEF cells at approximately physiological levels to assess the effects on cell function^{3,7}. To justify this approach with the DAFS residue variants, we showed that the focal adhesion (FA) morphometrics of Vcn-Venus, VcnCSs, and VcnTSs harboring DAFS residue variants were statistically identical (Fig.5, Fig.S7, Fig.S11). The manuscript has been edited to motivate these experiments more clearly.

We also wish to emphasize that Vcn^{-/-} MEFs were stably reconstituted with Vcn variants 1) at matched levels and 2) a level that we have previously shown to be consistent with the amount of Vcn expressed in litter-matched wild type MEFs^{3,6,8}. Thus, overexpression is not a major concern for these experiments.

We agree with the reviewer that the manuscript would benefit from further demonstration of the effects of DAFS residue variants on Vcn. However, assays involving externally applied forces could be hard to interpret due to changes in both Vcn exchange dynamics and load bearing anticipated with the loss of the directional catch, as well as the unintended activation of mechanotransduction pathways. To more definitively show the differences between the constructs in a manner that is related to catch bonding, we have chosen to utilize our FRET-FRAP technique to assess the correlation between the tension experienced by a protein and its exchange dynamics^{3,8}. Previously we have used this approach to show that Vcn forms tension-stabilized interactions within FAs, interactions with F-actin are required for tension-sensitive dynamics, and perturbation of Vcn: talin interactions leads to emergence of a tension-destabilized state³. FRET-FRAP analysis between VcnTS and the VcnTS E1015A-E1021A in stably expressed cells analyzed at approximately physiological expression revealed that these mutations lead to a substantial increase in the mobile fraction as well as a loss of tension-stabilized dynamics (Fig.7). We also validated that the FAs selected for FRET-FRAP analysis were similar and indicative of population averages (Fig. S13) and demonstrated there are no other statistically significant correlations between FRAP parameters and Vcn tension (Fig. S14). Overall, these data are consistent with DAFS variants of Vcn lacking directional catch bonding in cellulo, confirming the central point of this manuscript.

→ **Action:**

- Text describing the purpose of F-actin: FA alignment, directed migration assays, and FA sub-class analysis (see section 2.2) has been added to the text (Page 9).

- Text clearly stating that this is not an over-expression system, but a reconstitution system, at near physiological level of Vcn expression, has been added to the text (see Creation of Stable Cell Lines in Methods).
- Figures describing FRET-FRAP analysis of VcnTS and VcnTS E1015A-E1021A providing further evidence that the constructs have different behaviors has been added to the manuscript (Fig.7, Fig. S13, Fig. S14).

Figure 7 | DAFS variant VcnTSs exhibit loss of Vcn force-stabilized dynamics.

(A-B) Representative $Vcn^{-/-}$ VcnTS and $Vcn^{-/-}$ VcnTS E1015A-E1021A FAs undergoing FRAP time lapse, scale bar: 2 μ m **(C)** FRAP recovery curves are shown for cells expressing VcnTS and VcnTS E1015A-E1021A ($n=25$, 22 from 11 independent experimental days) **(D)** Box-whisker plots of mobile fraction of FAs analyzed in (C), one-way ANOVA followed by post-hoc Steel Dwass Test ($p<0.05$) **(E)** Box whisker plots of FRAP recovery half-time of FAs analyzed in (C), one-way ANOVA followed by post-hoc Tukey test ($p=0.2459$) **(F)** Correlation between FRAP recovery half-time and FRET efficiency for VcnTS corresponds to the force-stabilized state, one-way ANOVA ($n = 25$, $p=0.007$) **(G)** Correlation between recovery half-time and FRET efficiency for VcnTS E1015A-E1021A corresponds to the force-independent state, one-way ANOVA ($n = 22$, $p=0.49$) **(H)** Regression slopes calculated for panel (F and G). A least-squares linear regression was fit to the FRAP half-time and FRET efficiency data for each construct. Bars represent slope, and error bars represent standard error of the regression slope. Star indicates slope is statistically significant from zero as detected by t-test followed by Bonferroni Correction ($*p<0.0125$), here for VcnTS. See Table S1 for a detailed listing of p -values.

Fig. S13. Stably expressed VcnTS and VcnTS double DAFS variant characteristics are conserved from transient expression.

(A) Western blot of stably expressed VcnTS and DAFS double variant VcnTS E1015A-E1021A cell lines, showing presence of variants does not affect VcnTS cellular expression ability. Anti-GFP used to label VcnTS (B-C) Representative acceptor intensity image for Vcn^{-/-} VcnTS expressing cell and DAFS residue variant Vcn^{-/-} VcnTS E1015A-E1021A expressing cell. Box-whisker plots of (D) cell area (one-way ANOVA paired with Tukey test, $p=0.4546$) (E) FA area (one-way ANOVA paired with Tukey test, $p=0.1113$) (F) FA Acceptor intensity (one-way ANOVA paired with Tukey test, $p=0.6497$) and (G) FRET efficiency (one-way ANOVA paired with Steel-Dwass test, $p=0.0002$) for VcnTS population ($n=25$) and VcnTS E1015A-E1021A population ($n=22$) selected for FRET-FRAP assay. Differing letters denote significance at $p<0.05$.

Fig. S14. Stably expressed VcnTS and VcnTS double DAFS variant dynamics and mobile fraction are independent of acceptor intensity.

(A) Correlation between recovery half-time and acceptor intensity for VcnTS ($n=25$, $p=0.91$) and (B) for VcnTS E1015A-E1021A ($n=22$, $p=0.58$) (C) Regression slopes calculated for FRAP half-time vs. Venus mean intensity at the single FA. Bars represent slope and error bars represent standard error of the regression slope. None of the slopes are statistically different from zero, as determined by a t-test comparing the regression slope to zero. (D) Correlation between mobile fraction and acceptor intensity for VcnTS ($n=25$, $p=0.85$) and (E) for VcnTS E1015A-E1021A ($n=22$, $p=0.34$) (F) Regression slopes calculated for mobile fraction vs. Venus mean intensity. A least-squares linear regression was fit to the FRAP mobile fraction and acceptor intensity data for each construct. Bars represent slope and error bars represent standard error of the regression slope. None of the slopes are statistically different from zero, as determined by a t-test comparing the regression slope to zero. (G) Correlation between mobile fraction and FRET efficiency for VcnTS ($n=25$, $p=0.16$) and (H) for VcnTS E1015A-E1021A ($n=22$, $p=0.75$) (I) Regression slopes calculated for mobile fraction vs. FRET efficiency at the single FA. Bars represent slope and error bars represent standard error of the regression slope. None of the slopes are statistically different from zero, as determined by a t-test comparing the regression slope to zero. See Table S1 for a detailed listing of p -values.

Issue 2.2, FLIM Imaging:

- **Reviewer Comment:** *“Ideally, authors should use FLIM to determine FRET efficiencies, which is much more reliable (PMID: 23203070 and PMID: 30865383) than sensitized emission. The accuracy in sensitized emission has strong limitations because of the bleedthrough parameters and corrections for chromatic (and magnification) aberrations that need to be corrected for.”*
- **Response:** We appreciate the reviewer’s suggestion. We would like to clarify that the measurements we report are indeed FRET efficiency measurements obtained using a sensitized emission method⁹. We are not conflating FRET efficiency and FRET index, as is often done in the literature. Performing FLIM-FRET microscopy would be another means of measuring FRET efficiency differences between the constructs. Indeed, in the first paper validating the VcnTS in 2010, we employed FLIM-FRET microscopy to ensure proper functionality of the sensor¹. In our subsequent work, we employed a variety of approaches to enable measurement of FRET efficiency and molecular tension with sensitized emission with high precision, reproducibility, and at large scales^{2,6,7,10}. We find that our approaches do require more corrections and calibration factors than FLIM-FRET, but can determine FRET efficiency with greater spatial resolution and can more readily be combined with other approaches and techniques¹⁰. For instance, we used sensitized emission-based measurement of FRET efficiency to explore the spatial distribution of Vcn tension within FAs³, combined it with FRAP to enable measurements of force-sensitive protein dynamics in living cells (see issue 2.1), and have recently leveraged the greater throughput of this approach to create FRET-based biosensor datasets (containing greater than 500,000 FAs) sufficient for application of machine learning based classification of sub-cellular structures⁷.

Additionally, we would like to note that the sensors suggested by the reviewer in the provided citations have a design that facilitates their use in FLIM-FRET measurements but affects the mechanical response. FLIM-FRET (in the time domain) requires accurate fitting of a decay function that is related to the key fluorescent lifetimes in the system. The more lifetimes, the more challenging data analysis becomes. The sensors used in the citations given by the reviewer were engineered to function as mechanical switches and undergo a large conformation change at a specific force^{11,12}. This limits the number of states the sensor exists in and lowers the statistical power (i.e., photon counts) required for high quality FLIM measurements. However, the sensors are not sensitive to a broad range of forces, and essentially function as high/low FRET detectors. As the changes in mechanical loads in response to removing catch bonding activity were completely unknown before this study, we chose to use sensors that responded to a broad range of forces but perform better in sensitized emission-based approaches. This justifies our choice to use sensitized emission in these studies.

We leveraged the greater spatial resolution, higher throughput, and compatibility with machine learning-based classification algorithms to assess the ability of DAFS to affect Vcn loading in various classes of FAs. Specifically, we focused on immature, protrusion (PR)-associated FAs as well as mature, stress fiber (SF)-associated FAs. We observe that the disruption of directional catch bond formation reduces the degree of Vcn loading in both classes of FAs but perturbs the spatial organization of the load in stress fiber-associated FAs only (Fig.S15). This is consistent with our analyses showing that Vcn directional catch binding is required for proper co-regulation of FA and stress fibers (Fig. 8).

Action:

- We added a statement that the required flat-fielding, bleedthrough parameters, sub-pixel channel registration, and determination of microscope “calibration” parameters (G and k) were performed to ensure that FRET efficiency is calculated accurately (see Methods Section, Quantitative FRET Efficiency Measurements from Sensitized Emission).
- We added a statement that sensitized emission FRET efficiency measurements enable the FRET-FRAP analyses (page 9, see Issue 2.1 for further details) as well as study of the spatial variation and FA-class dependent effects of directional catch bonding (page 10).
- We added a supplemental figure depicting the role of directional catch bonding in mediating the magnitude and spatial organization of Vcn loading in various classes of FAs, establishing a key role for

directional catch bonding in mature, stress fiber-associated FAs (Fig S15) and further substantiating the analyses on FA and stress fiber co-alignment (Fig. 8).

Fig. S15. VcnTS DAFS double variant SF-associated FAs exhibit loss of characteristic load gradients.

(A) Representative images of cells stably expressing VcnTS or VcnTS E1015A-E1021A with machine learning-identified SF-associated or PR-associated FAs **(B)** Corresponding FRET image of cells from panel A, with pictured insets of SF or PR-associated FAs. FA Line scans of normalized acceptor intensity over normalized length of FA for representative **(C)** VcnTS SF-associated FAs (n=46) **(D)** VcnTS E1015A-E1021A SF-associated FAs (n=73) **(E)** VcnTS PR-associated FAs (n=63) **(F)** VcnTS E1015A-E1021A PR-associated FAs (n=67). FA line scans of FRET efficiency over normalized length of FA for **(G)** VcnTS SF-associated FAs (n=46) **(H)** VcnTS E1015A-E1021A SF-associated (n=73) FAs **(I)** VcnTS PR-associated FAs (n=63) **(J)** VcnTS E1015A-E1021A PR-associated FAs (n=67) **(K)** FA area for representative FAs in (C-J). Differing letters denote significance $p < 0.05$, one-way ANOVA paired with post-hoc Steel Dwass test. **(L)** Mean FA FRET efficiency for FAs in (C-J). Differing letters denote significance at $p < 0.05$, one-way ANOVA paired with Steel-Dwass Test. **(M)** Change in FRET efficiency across FA line scan for FAs in (C-J). Differing letters denote significance at $p < 0.05$, one-way ANOVA paired with Steel-Dwass Test. **(N)** Cell averaged FA area in pixels of SF- and PR- associated FAs of all cells expressing VcnTS or VcnTS E1015A-E1021A respectively (VcnTS SF FAs from n=102 cells, VcnTS E1015A-E1021A SF FAs from n=79 cells, VcnTS PR FAs from n= 103 cells, VcnTS E1015A-E1021A PR FAs from n= 79 cells). Differing letters denote significance at $p < 0.05$, one-way ANOVA paired with Steel-Dwass Test. **(O)** Cell averaged FA FRET efficiency of SF- and PR-associated FAs of cells from (N). Differing letters denote significance at $p < 0.05$, one-way ANOVA paired with Steel-Dwass Test. **(P)** Cell averaged spatial variation index of SF and PR associated FAs from (N). Differing letters denote significance at $p < 0.05$, one-way ANOVA paired with Steel-Dwass Test. See Table S1 for a detailed listing of p -values.

Issue 2.3, Format for presenting images:

- **Reviewer Comment:** “Whilst I like the thorough quantification of the sensor experiments, it would be better to show the complete image for each FRET construct, i.e., FAs together with the cytoplasm to see the relation between the two. Separate images can show the masks used for measurements and/or show the images for the separate areas.”
- **Response:** We thank the reviewer for appreciating our detailed analysis and for this suggestion. We have used the suggested format before³. Thus, we have altered the supplemental figures describing the segmentation process to match the reviewer’s suggestions. Additionally, in these figures we have added histograms of the FRET pixels values corresponding to the various subcellular compartments of representative images of VcnCS or VcnTS expressing cells. These clearly demonstrate that the model of the distributions in the cytosol match the estimates of closed/inactivated Vcn and unloaded VcnTS.
- **Action:**
- To demonstrate the segmentation analyses more clearly for VcnCS and VcnTS, Fig. S9 and Fig. S12 have been added to the manuscript.

Fig. S9. VcnCS and VcnCS DAFS variants are closed in the cytosol. (A) Representative acceptor image of Vcn^{-/-} VcnCS expressing MEF (B) Segmented FA mask image (C) Representative cytoplasm mask image generated from dilated FA mask (D) Representative image of FRET pixels in cell cytoplasm and FAs (E) Representative image of FA FRET pixels, from FA mask applied to FRET (F) Representative cytoplasmic FRET, from cytoplasm mask applied to cell (G-I) Histogram of FRET pixels in D-F (J) Box-whisker plots are shown for FRET efficiency of cytoplasmic VcnCS and DAFS variants of VcnCS, and VcnCS on pLL control ($n = 47, 50, 53, 63, 62$ cells, respectively from 3 independent experimental days). One-way ANOVA paired with a Steel-Dwass post-hoc test were performed for statistical analysis. Different letters denote significant difference at $p < 0.05$. See Table S1 for a detailed listing of p -values.

Fig. S12. VcnTS and VcnTS DAFS variants are unloaded in the cytosol.

(A) Representative acceptor image of $Vcn^{-/-}$ VcnTS expressing MEF (B) Segmented FA mask image (C) Representative cytoplasm mask image generated from dilated FA mask (D) Representative image of FRET pixels in cell cytoplasm and FAs (E) Representative image of FA FRET pixels, from FA mask applied to FRET (F) Representative cytoplasmic FRET, from cytoplasm mask applied to cell (G-I) Histogram of FRET pixels in D-F (J) Box-whisker plots are shown for FRET efficiency of cytoplasmic VcnTS, DAFS residue variants of VcnTS and VcnTS I997A ($n = 53, 68, 63, 80, 44$ cells, respectively). One-way ANOVA paired with Tukey post-hoc test was performed for statistical analysis. Different letters denote significant difference at $p < 0.05$. See Table S1 for a detailed listing of p -values.

Issue 2.4, Apparent gradients in FRET Efficiency Images:

→ **Reviewer Comment:** “All the images for quantification of the cytoplasmic pool of VcnCS (Fig.S5) and VcnTS sensors (Fig.S8) show a steep gradient of FRET decrease towards the cell edge. This does raise some concerns and one wonders whether the sensors do what they claim to do. What is the reason for this gradient? Were these measurements done in fixed or live cells?”

→ **Response:** We thank the reviewer for providing such a detailed review of our data. Indeed, the FRET efficiency values from some regions of the cytoplasmic portion of cell interior are quite high. This is due to relatively low levels of sensors in these regions. This differential localization is expected biologically, as Vcn has a strong preference for localizing to FAs at physiological expression levels^{3,13}. These low levels of expression lead to low intensities that increase the noise in the FRET calculations as well as lead to apparent non-linearities in the measurements of the bleedthrough¹⁴. We have included examples of bleedthrough coefficient curves below so that the reviewer can see the intensity levels that are affected. Others have suggested that these effects are due to the inherent non-linearities in photon detection in most cameras/detectors¹¹.

Bleedthrough Curves for Donor and Acceptor Channel

(A) Bleedthrough coefficient values for acceptor pixel intensity range detected by camera (0-60000), fitted to a polynomial. Linear bleedthrough of 0.24 observed at acceptor intensity above 1500 **(B)** Bleedthrough coefficient values for donor pixel intensity range detected by camera (0-60000), fitted to a polynomial. Linear bleedthrough of 0.75 observed at acceptor intensity above 1000.

As we only evaluate the cytoplasm as a control, we prefer to leave these data in the associated analysis. Furthermore, the dependence on brightness is the reason that we 1) segment images and show FRET efficiency signals for the FA and 2) use situations where high cytoplasmic expression can be achieved (such as the expression of TSMoD or plating cells on poly-L-lysine) as additional unloaded controls in other portions of the manuscript. We note that an analogous issue occurs in FLIM-FRET imaging, as insufficient photon counts are typically observed in the cytoplasm⁵.

→ **Action:**

- We provided images in the figures that more readily show differences between FRET efficiencies in the FA and cytoplasm in Fig.S9 and Fig. S12.
- We have added images of acceptor intensity, which is proportional to sensor concentration, to all figures displaying data from FRET-based biosensors (Fig. S8, Fig. S9, Fig. S10, Fig. 6, Fig. S12, and Fig. S15).
- We have included, just in this response letter, examples of the acceptor and donor bleedthrough curves that highlight the issues at low intensities. These data can be added to the manuscript, if the reviewer feels it would improve the manuscript.

Figure 6 | DAFS variant VcnTSs experience reduced, but not negligible load.

(A) Schematic of TSMOD and VcnTS constructs in loaded and unloaded states. **(B)** Schematic of VcnTS in the FA, loaded by actomyosin contractility. **(C-G)** FA masked acceptor intensity images are shown for single *Vcn*^{-/-} MEFs expressing WT VcnTS, DAFS variants of VcnTS and a previously validated variant with reduced actin affinity, VcnTS I997A ($n = 232, 85, 79, 97, 58$ respectively, pooled from 11 days). **(H)** Representative cytoplasm acceptor intensity shown for a single *Vcn*^{-/-} MEF expressing unloaded FRET control, TSMOD ($n = 69$, from individual experimental 5 days). **(I-M)** Corresponding FA masked FRET efficiency shown for (C-G). **(N)** Corresponding cytoplasm FRET efficiency shown in (H). **(O)** Box-whisker plots are shown for cell-averaged FRET efficiency of VcnTS and VcnTS DAFS variants compared to expected unloaded FRET efficiency (dotted line) controls. One-way ANOVA paired with a Steel-Dwass test was used for statistical analysis. Differing letters denote significant difference at $p < 0.05$. See Table S1 for a detailed listing of p -values.

Issue 2.5, Controls for VcnCS sensor

- **Reviewer Comment:** “There is no negative or positive control for the VcnCS sensor, and therefore it is unclear whether it works. Mutations published for a constitutively active Vcn (T12) would help.”
- **Response:** We apologize for not presenting the VcnCS data in a clear way. The VcnCS sensor was first created in 2005 and is widely used by us and others^{15,16,17}. There is a positive control (high FRET) control for VcnCS present in the paper (Fig.S8). We plated cells on poly-l-lysine treated glass coverslips, so that they do not spread and do not form FAs, preventing the activation of Vcn. This enables a high concentration of sensor in the cytoplasm and accurate determination of the closed state of the sensor. The T12 variant contains 4 mutations and there have been reports that it perturbs Vcn function¹⁸. In previous work, we have also identified mutations that perturb the head-tail inhibition within Vcn that require fewer mutations¹⁹. In previous work, we demonstrated that including these mutations in a VcnCS leads to the expected reduction in FRET¹⁹. Therefore, we are quite confident that VcnCS is working as expected.
- **Action:**
- We have altered the text to more clearly point readers to the positive control for VcnCS, and noted that experiments with cells plated on poly-l-lysine (Fig. S8) were used to generate a “closed FRET Efficiency for VcnCS” that was added to other figures involving VcnCSs (Fig. S9 and Fig S10).

Fig. S8. WT VcnCS expressing cells plated on poly-L-lysine coated glass show closed VcnCS FRET efficiency.

(A) Acceptor intensity and (B) FRET efficiency image of Vcn^{-/-} MEF expressing VcnCS, which localizes diffusely in the cytosol when plated on pLL-coated glass (C) Schematic depicting closed conformation (high FRET) and open conformation (low FRET) of VcnCS in the cytosol (D) Box-whisker plot of FRET efficiency ($n = 62$ cells, respectively, from 3 independent experimental days).

Fig. S10. WT VcnCS and VcnCS DAFS variants maintain open conformation at FAs.

(A-D) FA acceptor intensity and (E-H) FA masked FRET efficiency is shown for representative *Vcn*^{-/-} MEFs expressing WT VcnCS and DAFS variant VcnCS constructs. (I) Box-whisker plots are shown for cell-averaged FA FRET efficiency of VcnCS and VcnCS DAFS variants ($n = 76, 94, 99, 109$ cells, respectively, from 5 independent experimental days) compared to closed VcnCS FRET efficiency (dotted line). One-way ANOVA paired with Tukey's HSD test was used for statistical analysis. Different letters denote significant difference at $p < 0.05$. See Table S1 for a detailed listing of p -values.

Issue 2.6, Comparison of TSMoD and Cytosolic Signals:

- Reviewer Comment: *“The TSMoD seems rather low in FRET (at least the image) in comparison with the cytoplasmic fraction of the various TS sensors (Fig.S8). Were they differently scaled or are the differences due to differences in the rotational freedom or other properties of the probes?”*
- **Response:** We thank the reviewer again for providing such a detailed review of our data. We will first discuss the possibility of differences in rotational freedom of fluorescent protein or other properties of the probes affecting their function. In the case of the tension sensors, it was shown that linking two fluorescent proteins by flexible linkers of this size does not affect their rotational freedom, as measured by fluorescence anisotropy²⁰. Then, we showed that the piece-wise addition of Vcn domains to a tension sensitive module did not affect FRET in solution¹. Furthermore, a single point mutation that prevents actin binding or the use of actomyosin inhibitors can be used to return VcnTS to FRET efficiencies consistent with those observed in the soluble TSMoD³. Additionally, these findings have been generalized to a variety of FRET-based tension sensors^{13,21-25}. We have also determined that the Forster equation involving slow diffusion of a fluorophore (e.g., a constant orientation for any given FRET event, but a random distribution of orientation over all FRET events) provides a better description of tension sensor behavior compared to the standard Forster equation assuming dynamic isotropic regime for fluorescent moiety rotation (i.e. $\kappa^2 = 2/3$ approximation)². This slower mobility for fluorescent proteins, which are much larger than organic dyes often used in FRET, is expected. Thus, there is no evidence that rotational freedom of the fluorescent proteins is impeded in the tension sensors, and we do not consider this a primary concern. For VcnCS, specific mutations and exposure to bioactive ligands (e.g., IpaA) have been used to show that relief of the head-tail interaction within Vcn leads to a specific loss of FRET in this sensor^{1,15}. Notably the internal FP of VcnCS was inserted in the same position as VcnTS, suggesting the more rigorous testing done for VcnTS is applicable. Together, these data suggest that mobility of FPs in VcnCS is also not impeded.

Similar to Issue 2.4, the discrepancy the reviewer is noticing is due to the difference in intensity (i.e., the relative amount) of sensor that can be achieved in cytoplasm in experiments involving VcnTS and TSMoD (compare Fig. 6H and Fig 12A). Endogenous expression levels of Vcn results in strong localization to the FAs and a substantially smaller pool in the cytoplasm¹³. In contrast, TSMoD is a soluble protein and strongly localizes to the cytoplasm. Thus, intensity levels of TSMoD in the cytoplasm are comparable to those observed with VcnTS and VcnCS in FAs, facilitating direct comparisons. We note that FRET efficiency of TSMoD and that of VcnTS I997A within FAs are not statistically different and consistent with previously determined estimates and theoretical predictions of the unloaded FRET efficiency². Based on these data and our previous experience, we conclude that the sensors used in the manuscript are functioning as expected.

→ Action:

- We have added a sentence to the main text that the sensors have previously been evaluated for artifacts associated with rotational differences, and none were observed (page 8).
- As described in Issue 2.4, all figures involving FRET-based biosensors have been altered to include images from the acceptor channel, which are indicative of sensor localization (Fig. S8, Fig. S9, Fig. S10, Fig. 6, Fig. S12, and Fig. S15).

- 1 Grashoff, C. *et al.* Measuring mechanical tension across vinculin reveals regulation of focal adhesion dynamics. *Nature* **466**, 263-266, doi:10.1038/nature09198 (2010).
- 2 LaCroix, A. S., Lynch, A. D., Berginski, M. E. & Hoffman, B. D. Tunable molecular tension sensors reveal extension-based control of vinculin loading. *eLife* **7**, e33927, doi:10.7554/eLife.33927 (2018).
- 3 Rothenberg, K. E., Scott, D. W., Christoforou, N. & Hoffman, B. D. Vinculin force-sensitive dynamics at focal adhesions enable effective directed cell migration. *Biophysical Journal* **114**, 1680-1694 (2018).
- 4 Chang, C.-W. & Kumar, S. Vinculin tension distributions of individual stress fibers within cell–matrix adhesions. *Journal of Cell Science* **126**, 3021-3030, doi:10.1242/jcs.119032 (2013).
- 5 Li, F. *et al.* Vinculin force sensor detects tumor-osteocyte interactions. *Scientific reports* **9**, 1-11 (2019).
- 6 Rothenberg, K. E., Neibart, S. S., LaCroix, A. S. & Hoffman, B. D. Controlling cell geometry affects the spatial distribution of load across vinculin. *Cellular and Molecular Bioengineering* **8**, 364-382 (2015).
- 7 Tao, A. *et al.* Identifying constitutive and context-specific molecular-tension-sensitive protein recruitment within focal adhesions. *Dev Cell* **58**, 522-534.e527, doi:10.1016/j.devcel.2023.02.015 (2023).
- 8 Rothenberg, K. E., Puranam I., Hoffman Brenton D. . Measurement of Force-Sensitive Protein Dynamics in Living Cells Using a Combination of Fluorescent Techniques. *JoVE Journal* doi:10.3791/58619 (2018).
- 9 Chen, H., Puhl, H. L., Koushik, S. V., Vogel, S. S. & Ikeda, S. R. Measurement of FRET Efficiency and Ratio of Donor to Acceptor Concentration in Living Cells. *Biophysical Journal* **91**, L39-L41, doi:<https://doi.org/10.1529/biophysj.106.088773> (2006).
- 10 Gates, E. M., LaCroix, A. S., Rothenberg, K. E. & Hoffman, B. D. Improving Quality, Reproducibility, and Usability of FRET-Based Tension Sensors. *Cytometry Part A* **95**, 201-213, doi:<https://doi.org/10.1002/cyto.a.23688> (2019).
- 11 Wallrabe, H. & Periasamy, A. Imaging protein molecules using FRET and FLIM microscopy. *Current Opinion in Biotechnology* **16**, 19-27, doi:<https://doi.org/10.1016/j.copbio.2004.12.002> (2005).
- 12 Sun, Y., Rombola, C., Jyothikumar, V. & Periasamy, A. Förster resonance energy transfer microscopy and spectroscopy for localizing protein–protein interactions in living cells. *Cytometry Part A* **83**, 780-793, doi:<https://doi.org/10.1002/cyto.a.22321> (2013).
- 13 Humphries, J. D. *et al.* Vinculin controls focal adhesion formation by direct interactions with talin and actin. *Journal of Cell Biology* **179**, 1043-1057, doi:10.1083/jcb.200703036 (2007).
- 14 Chen, Y. E., Mauldin, J. P., Day, R. N. & Periasamy, A. Characterization of spectral FRET imaging microscopy for monitoring nuclear protein interactions. *Journal of Microscopy* **228**, 139-152, doi:<https://doi.org/10.1111/j.1365-2818.2007.01838.x> (2007).
- 15 Chen, H., Cohen, D. M., Choudhury, D. M., Kioka, N. & Craig, S. W. Spatial distribution and functional significance of activated vinculin in living cells. *J Cell Biol* **169**, 459-470, doi:10.1083/jcb.200410100 (2005).
- 16 Cohen, D. M., Kutscher, B., Chen, H., Murphy, D. B. & Craig, S. W. A Conformational Switch in Vinculin Drives Formation and Dynamics of a Talin-Vinculin Complex at Focal Adhesions ^{*}. *Journal of Biological Chemistry* **281**, 16006-16015, doi:10.1074/jbc.M600738200 (2006).
- 17 Cohen, D. M., Chen, H., Johnson, R. P., Choudhury, B. & Craig, S. W. Two distinct head-tail interfaces cooperate to suppress activation of vinculin by talin. *Journal of Biological Chemistry* **280**, 17109-17117 (2005).
- 18 Chorev, D. S. *et al.* Conformational states during vinculin unlocking differentially regulate focal adhesion properties. *Sci Rep* **8**, 2693, doi:10.1038/s41598-018-21006-8 (2018).
- 19 Thievensen, I. *et al.* Vinculin–actin interaction couples actin retrograde flow to focal adhesions, but is dispensable for focal adhesion growth. *Journal of Cell Biology* **202**, 163-177, doi:10.1083/jcb.201303129 (2013).
- 20 Evers, T. H., van Dongen, E. M., Faesen, A. C., Meijer, E. W. & Merckx, M. Quantitative understanding of the energy transfer between fluorescent proteins connected via flexible peptide linkers. *Biochemistry* **45**, 13183-13192, doi:10.1021/bi061288t (2006).
- 21 Kumar, A. *et al.* Talin tension sensor reveals novel features of focal adhesion force transmission and mechanosensitivity. *J Cell Biol* **213**, 371-383, doi:10.1083/jcb.201510012 (2016).
- 22 Borghi, N. *et al.* E-cadherin is under constitutive actomyosin-generated tension that is increased at cell–cell contacts upon externally applied stretch. *Proceedings of the National Academy of Sciences* **109**, 12568-12573, doi:doi:10.1073/pnas.1204390109 (2012).
- 23 Ryan, G. H. *et al.* Myosin II Tension Sensors Visualize Force Generation within the Actin Cytoskeleton in Living Cells. *bioRxiv*, 623249, doi:10.1101/623249 (2019).
- 24 Kanoldt, V. *et al.* Metavinculin modulates force transduction in cell adhesion sites. *Nature Communications* **11**, 6403, doi:10.1038/s41467-020-20125-z (2020).

25 Kim, T. J. *et al.* Dynamic visualization of α -catenin reveals rapid, reversible conformation switching between tension states. *Curr Biol* **25**, 218-224, doi:10.1016/j.cub.2014.11.017 (2015).

REVIEWER COMMENTS

Reviewer #1 (Remarks to the Author):

In the manuscript titled "Elucidation of the Molecular Basis and Cellular Functions of Vinculin-Actin Directional Catch Bonding," Chirasani et al. discuss the intriguing phenomenon of cells and tissues responding differently to mechanical forces based on the direction of application. The authors explore this responsiveness through specific load-bearing proteins capable of maintaining preferential physical linkages in certain directions. The focus of the paper is on vinculin (Vcn), a load-bearing linker protein that exhibits directional catch bonding, where the strength of a bond between two molecules increases under mechanical force, contrary to typical slip bonds.

The researchers present a computational approach to predict the specific Vcn residues involved in directional catch bonding. Additionally, they create a set of Vcn variants that retain the original Vcn tail domain (Vt) structure, actin binding capability, and interactions with phospholipids. Although incorporating these variants into Vcn does not affect its activation, it leads to reduced Vcn loading and altered Vcn exchange dynamics, indicating a potential loss of directional catch bonding. The authors further demonstrate that the expression of these Vcn variants significantly affects the coordination of subcellular structures and cell migration, highlighting the crucial role of directional catch bonding in key cellular functions related to cell movement and organization.

While the work is interesting, there are several areas that need improvement before I can recommend it for publication in Nature Communications.

Bibliographic Advancements: The manuscript lacks acknowledgment and discussion of recent advances in the study of mechanical properties of proteins and protein complexes. Over the past decade, numerous works have investigated directionality-dependent bonds, including catch-bonds, through techniques like Molecular Dynamics simulations in combination with single molecule force spectroscopy experiments. It would be valuable to place the current findings in the context of these published studies, which have shown similar phenomena in other protein complexes. The authors should consider citing relevant works from the bibliography to strengthen the manuscript's discussion. Also, the authors should tone-down some of the statements as the methodology presented here, particularly for the computational work, is not exactly novel.

Consideration of Mechanical vs. Thermodynamic Binding Affinity: The authors should address the decorrelation between mechanical stability and thermodynamic binding affinity in non-covalent protein interactions. While early studies correlated mechanical stability with binding affinity in the absence of force, it is now widely recognized that these two aspects can be separate. Catch bonds and matrix-adhesive receptors optimized for high mechanostability through evolution are examples of such interactions. I have reservations about whether the umbrella sampling method used in this manuscript, or the suite for DAFS Vt variants, can fully capture this nuance in the mechanical vs. thermodynamic binding affinity. The authors should discuss this potential limitation and explore alternative approaches to investigate this aspect.

Exploring Other Interaction Mechanisms: The manuscript primarily focuses on the hydrogen bond network, but it is essential to consider other possible types of interactions involved in catch-bonding. Many catch-bonds operate through different mechanisms or even a combination of mechanisms. The authors should expand their discussion to encompass the possibility of multiple interaction types contributing to directional catch bonding in the vinculin-actin system.

Addressing these points will significantly enhance the manuscript's clarity and impact, making it more suitable for publication in Nature Communications.

Reviewer #3 (Remarks to the Author):

I am happy with the comments/changes and believe that the manuscript has significantly improved.

REVIEWER 1 COMMENTS:

Issue 1.1, Acknowledgment and discussion of recent advances in the study of directionality-dependent interactions in protein complexes:

→ **Reviewer Comment:** *The manuscript lacks acknowledgment and discussion of recent advances in the study of mechanical properties of proteins and protein complexes. Over the past decade, numerous works have investigated directionality-dependent bonds, including catch-bonds, through techniques like Molecular Dynamics simulations in combination with single molecule force spectroscopy experiments. It would be valuable to place the current findings in the context of these published studies, which have shown similar phenomena in other protein complexes. The authors should consider citing relevant works from the bibliography to strengthen the manuscript's discussion. Also, the authors should tone-down some of the statements as the methodology presented here, particularly for the computational work, is not exactly novel.*

→ **Response:** This comment has two key points: 1) the proper contextualization of the manuscript and 2) the removal of statements claiming novelty of the overall methodology, particularly in reference to the computational methodology. We will address the contextualization issue first. The edited text in all our responses below is shown in red for your convenience.

We appreciate the reviewer's suggestion to further contextualize our findings with recent advances in studies of protein mechanical properties. We agree with the review that this field is large with many recent advances and note that it can be challenging to summarize this field succinctly. We had previously limited our focus to mechanical linker proteins found within the adhesive structures of mammalian cells (e.g., focal adhesion and adherens junctions) (see Fig. S1), as this is what our biological experiments are focused on. Relevant works utilizing Molecular Dynamics^{1,2} and SMFS studies³⁻⁸ in this area were cited. Moreover, the reference number limit for the journal is 70. Of note, there are relatively few studies like ours that integrate computational, biochemical, biophysical and cell biology approaches within the sub-fields of cell-cell and cell-ECM biology. To our knowledge, the only paper is one focusing on cadherin¹. We have cited this paper and utilize it as the basis through which we define H-bonds.

Based on the reviewer's suggestion we have expanded the discussion section to include recent developments from a broader set of systems, in particular focusing on bacterial systems exposed to fluid shear stresses. We had originally not focused on these studies given the reference citation limit for the journal, also because the key forces are much larger and changes in protein structure associated with catch bonding are much more substantial. We now place our results in the context of this work within the discussion. Specifically, we note that the forces involved in catch bonding are much higher and the protein conformation changes are more substantial in the context of bacterial adhesion. Comparing these various systems will likely be a fruitful endeavor in the future.

We acknowledge your observation that the methodology presented, particularly the computational aspect, is not entirely novel. However, to our knowledge, this is the first study on directional catch bonding that presents an integrated approach, encompassing the identification of key amino acids through *in silico* analyses, followed by functional validation of mutations using *in vitro* biochemical and biophysical assays, assessment, and exploration of the impact on cellular processes using *in cellulo* biosensors. We revised the Introduction section and several other portions of the manuscript (see Action for details) to ensure that our contributions are presented in a suitable perspective while acknowledging the established computational methodologies in the field.

→ **Action:** We added the following paragraph to the discussion section and acknowledged recent advances in the study of mechanical properties of proteins and protein complexes.

Actions Re: Improving Contextualization

We have revised the following statements to enhance their clarity.

Page 13: “Overall, this work develops and implements an interdisciplinary approach for identifying, perturbing, and assessing key biological functions of directional catch bonding between F-actin and Vcn. In this work we focused on cell–ECM adhesion and single cell migration. Vcn also has key roles in cell–cell adhesion, collective cell migration, and a variety of developmental as well as pathophysiological mechanosensitive processes where directional catch bonding is likely to play key, but undefined, roles. These can now be explored with the tools and approaches developed here. Furthermore, this approach will likely be generalizable to other F-actin binding, loading-bearing proteins and will enable assessment and potential manipulation of directional catch bonding in this large class of proteins. and mediate diverse processes, such as binding of neutrophils to inflamed endothelial cells⁹ and T-cell receptor activation⁶². **Directional catch bonds also play a key role in bacterial adhesion, but the relevant forces and protein conformations are much larger and more substantial than those involved in mammalian cell adhesion⁶²⁻⁶⁴. An important question for future work will be determining if similar or distinct mechanisms mediate directional catch bonding in these mechanically-distinct regimes. Overall, we expect that the integrated procedure for identifying key residues and producing tools to probe the effects of the loss of directional catch bonding developed here will facilitate a plethora of studies in mechanobiology at the molecular, cellular, and tissue level across these wide-ranging fields.**”

Actions Re: Potential Novelty of Approaches

We modified following statements for clarity.

Page 2: “The ability of **mammalian** cells to generate, resist, and respond to mechanical forces applied in distinct directions is critical to many fundamental biological processes, including the maintenance of cytoskeletal order, directed cell migration, and the development of anisotropic mechanical properties of tissue¹⁻⁴.”

Page 3: “Umbrella sampling revealed that this simplified system captured key aspects of directional asymmetric **Vcn:F-actin** catch bond previously observed using single molecule approaches.”

Page 4: “Notably, these computational results correlate well with SMFS studies probing the relative strength of directional catch bonding between Vt and F-actin, validating the overall computational approach for studying **this interaction**.”

Page 11: “The ability of **mammalian** cells and tissues to resist or deform differentially in response to mechanical forces applied in distinct directions is mediated by the ability of load-bearing proteins to maintain or change connectivity in response to these forces^{52,53}. **In the context of cell-ECM adhesion**, previous work has identified several key examples, and simulations have predicted key roles for directional catch bonds in polar or directed molecular and cellular process^{11,17}. To test these predictions, we **developed an integrated procedure enabling the *in silico* identification, rational engineering, biochemical validation, and probing of the *in cellulo* roles of amino acids mediating directional catch bonding.**”

Page 11: “Our initial analysis of the umbrella sampling simulations provided valuable insights into the energetics and stability of the Vt:F-actin complex under different pulling directions **and provided a key connection to experimental observations of directional catch bonding¹¹.**”

Issue 1.2, Consideration of Mechanical vs. Thermodynamic Binding Affinity:

- **Reviewer Comment:** *“The authors should address the decorrelation between mechanical stability and thermodynamic binding affinity in non-covalent protein interactions. While early studies correlated mechanical stability with binding affinity in the absence of force, it is now widely recognized that these two aspects can be separate. Catch bonds and matrix-adhesive receptors optimized for high mechanostability through evolution are examples of such interactions. I have reservations about whether the umbrella sampling method used in this manuscript, or the suite for DAFS Vt variants, can fully capture this nuance in the mechanical vs. thermodynamic*

binding affinity. The authors should discuss this potential limitation and explore alternative approaches to investigate this aspect.”

Response: We agree with the reviewer’s observation that the decorrelation between mechanical stability and thermodynamic binding affinity has been a major advancement in the study force-sensitive behaviors of deformable proteins (i.e. ssDNA) or very high force scenarios (bacterial adhesion)^{10–13}. We also agree with the reviewer that exploring the effects of biopolymer mechanics, protein deformability, and conformational changes on force-sensitive binding kinetics is intriguing and holds great potential for future investigations. A particularly interesting aspect is the elucidation of distinct low-force and high unfolding pathways¹³. However, our study primarily aims to elucidate the directional catch bonding mechanism and associated molecular interactions between F-actin and Vcn with eventual extension to other force-bearing proteins within focal adhesions and adherens junctions. In these sub-cellular structures, the pertinent biopolymers and proteins are rather stiff, and the relevant forces are rather low. Furthermore, we primarily use the umbrella sampling to make a touch point to experiments demonstrating the directional catch bonding of the Vt:F-actin complex⁴, not to predict the overall mechanosensitivity of the complex. Given, these facts and limitations, we believe that we have employed the umbrella sampling technique properly and interpreted it appropriately. However, we agree with the reviewer’s concern on potential limitations of umbrella sampling simulations in the presence of large forced-induced conformational change (either when in biopolymers or binding proteins). We have added a supplemental note discussing the potential issue in the context of other systems.

➔ **Action:** We have included the following Supplemental Note on potential limitation of umbrella sampling, how we interpret the results, and potential alternative approaches that could be further explored in future work.

“In this work we used umbrella sampling to assess directional catch bonding energetics of the Vt:F-actin complex. This was done to enable a qualitative comparison to previous experimental work describing the directional dependence of Vt:F-actin catch bonding^{1,2}. It is worth mentioning here that while umbrella sampling is a valuable technique for exploring the energetic aspects of molecular interactions under force, it might not yield an exact quantification of binding affinity³. However, the approach is justified here as the relevant proteins (F-actin and Vt) are relatively stiff, the forces are low, and the conformation changes are small. These criteria may not be met in systems with soft biopolymers (i.e. single stranded DNA^{4,5}) where a plethora of conformation are expected or scenarios involving large force and substantial protein conformational changes (i.e. bacterial adhesion proteins^{6–8}). Furthermore, we have purposefully only made qualitative comparisons (e.g., the ordering of the bond strength as function pulling direction), as interpreting the simulation results quantitatively (e.g. as an indication of bond affinity) may represent an over-interpretation of the data. Several alternative methodologies, such as steered molecular dynamics (SMD)⁹ and enhanced sampling techniques like metadynamics¹⁰ or replica exchange simulations¹¹, may be required for such a detailed investigation of catch bonding energetics between Vt and F-actin.”

Issue 1.3, Exploring Other Interaction Mechanisms:

➔ **Reviewer Comment:** *“The manuscript primarily focuses on the hydrogen bond network, but it is essential to consider other possible types of interactions involved in catch-bonding. Many catch-bonds operate through different mechanisms or even a combination of mechanisms. The authors should expand their discussion to encompass the possibility of multiple interaction types contributing to directional catch bonding in the vinculin-actin system.”*

➔ **Response:** We appreciate the reviewer’s observation regarding the focus on the hydrogen bond network in our manuscript and that it is important to consider other possible interaction mechanisms involved in catch bonding. We note that in the previous iteration, we greatly expanded our analyses of the various types of intermolecular interactions. Please see Fig S2 which probes the importance of contact area, salt-bridges, and van der Waals forces associated with Vcn:F-actin catch bonds and Fig S5 A-C that highlights key roles for Vt E1015 and E1021 substitutions in reducing these intermolecular interactions. Based on the reviewer comments, we have now

expanded our discussion to highlight these data, describe a key physical principle observed (that force-induced formation of H-bonds is required for the reinforcement of other types of intermolecular interactions to mediate strong catch bonding), and note that a major question will be determining the generalization of this principle in other catch bonds. We have also created a table (Table S2) to list all salt-bridge and non-bonded interactions between Vt and F-actin in the F_{pointed} direction.

→ **Action:** We have altered the discussion to address the reviewer's concerns. The red colored font indicates changes made to the main text.

Page 11: "The subsequent DMD simulations of the Vt:F-actin complex in different directions relative to F-actin captured force-induced conformational changes and non-covalent interactions that contribute to catch bonding. H-bond occupancy analyses suggested that directional catch bonding between Vt and F-actin was driven by DAFS H-bond formation by Vt residues E1015 and E1021. In silico mutagenesis studies identified E1015A, E1021A, and E1015A-E1021A as single and double Vt variants that should ablate pertinent interactions without perturbing Vt structure. DMD simulations indicated disruption of key Vt DAFS H-bonds with F-actin by the E1015A-E1021A double variant. This variant was also predicted to enhance disruption of other intermolecular interactions when subjected to forces in the F_{pointed} direction, **establishing a key role of H-bonds formed by E1015 and E1021 in reinforcing a variety of other types of intermolecular interactions to facilitate strong catch bonding (Table S2). Altogether, these findings strongly indicate that residues E1015 and E1021 play a pivotal role as key mediators of Vcn's directional catch bonding with F-actin.**"

Page 12/13: "Based on our findings, we propose a working model of the mechanism of directional catch bonding between Vcn and F-actin, as illustrated in Fig. S19. Overall, catch bonding seems to be due to a combination of force-induced structural changes in Vt and the asymmetric nature of F-actin. Initially, when the pulling force is applied in the F_{pointed} direction, Vt strongly adheres to protomer-B of F-actin. This causes a reorientation of Vt along the short axis of F-actin as well as rearrangement of the helical bundles within Vt. These changes bring Vt closer to F-actin and expose the side chains of E1015 and E1021, enabling the formation of non-native H-bonds with R335 on protomer-B and R147 on protomer-P, respectively (Fig. S18). The H-bonds between E1015/E1021 in Vt and R335/R147 in F-actin create a unique binding interface that has an enhanced contact area and reinforces a variety of intermolecular interactions that enable the substantial strength on the Vcn catch bond when forces are applied in F_{pointed} direction. Notably, the twisting of Vt, rearrangement of helical bundles, and exposure of E1015 and E1021 are not observed when Vt is pulled in the F_{barbed} direction. **Therefore, forces in the F_{barbed} direction result in formation of distinctly weaker intermolecular interactions, as existing H-bonds are strengthened to a lesser degree and salt-bridges and hydrophobic interactions are not reinforced.** This directional asymmetry is likely due to the structural heterogeneity and intrinsic polarity of F-actin⁵⁷. (Fig. S19). **An important question for future work will be determining whether the reinforcement observed in other types of intermolecular interactions by forced-induced H-bond formation is a common scenario in directional catch bonding.**"

Additionally, we have inserted a new table (Table S2) to highlight the contributions of non-H-bond interactions in Vcn:F-actin catch bonding. We have also incorporated the following paragraph in the results section to elaborate on the collective contribution of various interaction types to directional catch bonding in the Vcn:F-actin system.

Page 5: "**Apart from H-bond interactions, we have identified several hydrophobic (non-bonded) and salt-bridge interactions that play a significant role in mediating directional catch bonding between F-actin and Vt in the F_{pointed} direction (Table S2). The reinforcement of these interactions, facilitated by key force-induced hydrogen bonds, contributes significantly to the establishment of unique catch bonding directionality. The combination of H-bond, salt-bridge, and non-bonded interactions is instrumental in establishing the unique directionality of catch bonding. Further, the salt-bridges and non-bonded interactions collectively enhance the stability of the Vt:F-actin complex under mechanical force. Our multi-faceted approach in the current study underscores the complexity of catch bonding, emphasizing the importance of considering a range of interaction types to comprehensively understand the directional specificity observed in the Vcn:F-actin system.**"

References:

1. Manibog, K., Li, H., Rakshit, S. & Sivasankar, S. Resolving the molecular mechanism of cadherin catch bond formation. *Nat Commun* **5**, 3941 (2014).
2. Thompson, P. M. *et al.* Identification of an actin binding surface on vinculin that mediates mechanical cell and focal adhesion properties. *Structure* **22**, 697–706 (2014).
3. Owen, L. M., Bax, N. A., Weis, W. I. & Dunn, A. R. The C-terminal actin-binding domain of talin forms an asymmetric catch bond with F-actin. *Proc Natl Acad Sci U S A* **119**, (2022).
4. Huang, D. L., Bax, N. A., Buckley, C. D., Weis, W. I. & Dunn, A. R. Vinculin forms a directionally asymmetric catch bond with F-actin. *Science (1979)* **357**, 703–706 (2017).
5. Buckley, C. D. *et al.* Cell adhesion. The minimal cadherin-catenin complex binds to actin filaments under force. *Science* **346**, (2014).
6. Kong, F., García, A. J., Mould, A. P., Humphries, M. J. & Zhu, C. Demonstration of catch bonds between an integrin and its ligand. *J Cell Biol* **185**, 1275–1284 (2009).
7. Del Rio, A. *et al.* Stretching single talin rod molecules activates vinculin binding. *Science* **323**, 638–641 (2009).
8. Guo, B. & Guilford, W. H. Mechanics of actomyosin bonds in different nucleotide states are tuned to muscle contraction. *Proc Natl Acad Sci U S A* **103**, 9844–9849 (2006).
9. Morikis, V. A. *et al.* Selectin catch-bonds mechanotransduce integrin activation and neutrophil arrest on inflamed endothelium under shear flow. *Blood* **130**, 2101–2110 (2017).
10. Liu, Z. *et al.* High force catch bond mechanism of bacterial adhesion in the human gut. *Nat Commun* **11**, 4321 (2020).
11. Bernardi, R. C. *et al.* Mechanisms of Nanonewton Mechanostability in a Protein Complex Revealed by Molecular Dynamics Simulations and Single-Molecule Force Spectroscopy. *J Am Chem Soc* **141**, 14752–14763 (2019).
12. Koirala, D., Yangyuoru, P. M. & Mao, H. Mechanical affinity as a new metrics to evaluate binding events. **32**, 197–208 (2013).
13. Melo, M. C. R., Gomes, D. E. B. & Bernardi, R. C. Molecular Origins of Force-Dependent Protein Complex Stabilization during Bacterial Infections. *J Am Chem Soc* **145**, 70–77 (2023).